# Structural insights into the iron nitrogenase complex

Frederik V. Schmidt [1,4], Luca Schulz [2,4], Jan Zarzycki [2], Simone Prinz[3], Niels N. Oehlmann[1], Tobias J. Erb [2] & Johannes G. Rebelein [1] ✉

Nitrogenases are best known for catalyzing the reduction of dinitrogen to ammonia at a complex metallic cofactor. Recently, nitrogenases were shown to reduce carbon dioxide ($CO_2$) and carbon monoxide to hydrocarbons, offering a pathway to recycle carbon waste into hydrocarbon products. Among the three nitrogenase isozymes, the iron nitrogenase has the highest wild-type activity for the reduction of $CO_2$, but the molecular architecture facilitating these activities has remained unknown. Here, we report a 2.35-Å cryogenic electron microscopy structure of the ADP·AlF$_3$-stabilized iron nitrogenase complex from *Rhodobacter capsulatus*, revealing an [Fe$_8$S$_9$C-(*R*)-homocitrate] cluster in the active site. The enzyme complex suggests that the iron nitrogenase G subunit is involved in cluster stabilization and substrate channeling and confers specificity between nitrogenase reductase and catalytic component proteins. Moreover, the structure highlights a different interface between the two catalytic halves of the iron and the molybdenum nitrogenase, potentially influencing the intrasubunit 'communication' and thus the nitrogenase mechanism.

Nitrogenases catalyze a key step in the global nitrogen cycle by reducing molecular nitrogen (dinitrogen, $N_2$) to ammonia ($NH_3$). Together with the energy-intensive industrial Haber–Bosch process, nitrogenases provide the vast majority of bioavailable nitrogen, which is essential for all life on Earth to build central metabolites such as nucleotides and amino acids[1–3]. Due to the extraordinary ability of nitrogenases to break the stable N≡N triple bond under ambient conditions, the mechanism of nitrogenases has been under scrutiny for decades[4–8].

To date, three homologous nitrogenase isoforms are known[9,10]. The most prevalent and best studied nitrogenase is the molybdenum (Mo) nitrogenase (encoded by *nifHDK*), which is present in all known diazotrophs[11]. Some diazotrophs encode 'back-up' or alternative nitrogenase genes: *vnfHDGK* for the vanadium (V) nitrogenase or *anfHDGK* for the iron (Fe) nitrogenase, expressed upon the depletion of Mo. All three nitrogenases consist of two components: the reductase component protein (NifH$_2$, VnfH$_2$ or AnfH$_2$) and the catalytic component protein (Nif(DK)$_2$, Vnf(DGK)$_2$ or Anf(DGK)$_2$). Importantly, the catalytic component of both alternative nitrogenases contains an additional subunit (VnfG or AnfG), whose role and function remain elusive[9,10].

The homodimeric reductase component contains an [Fe$_4$S$_4$] cluster and two adenosine triphosphate (ATP) binding sites. In the ATP-bound state, the reductase component transiently associates with its catalytic component. Upon complex formation, low-potential electrons are transferred from the [Fe$_4$S$_4$] cluster of the reductase component via an [Fe$_8$S$_7$] relay (P-cluster) to the active site cofactor of the catalytic component. The active site cofactor is expected to follow the general composition of [MFe$_7$S$_9$C-(*R*)-homocitrate], where M is Mo, V or Fe, depending on the nitrogenase isoform. Based on the containing heterometal, the clusters are termed FeMoco, FeVco or FeFeco. The structures of the FeMoco, the FeVco and just recently the FeFeco have been structurally confirmed by X-ray crystallography[12–16]. These studies revealed that in the FeVco, one of the belt-sulfur (S) atoms is replaced by a carbonate, resulting in a [VFe$_7$S$_8$C(CO$_3$)$^{2-}$)(*R*)-homocitrate] cluster. Based on the similar architecture of nitrogenases, one might expect

[1]Microbial Metalloenzymes Research Group, Max Planck Institute for Terrestrial Microbiology, Marburg, Germany. [2]Department of Biochemistry and Synthetic Metabolism, Max Planck Institute for Terrestrial Microbiology, Marburg, Germany. [3]Central Electron Microscopy Facility, Max Planck Institute of Biophysics, Frankfurt am Main, Germany. [4]These authors contributed equally: Frederik V. Schmidt, Luca Schulz. ✉e-mail: johannes.rebelein@mpi-marburg.mpg.de

them to follow a general catalytic mechanism. However, under 1 atm $N_2$ and high electron flux conditions (ratio of reductase to catalytic component ≥ 20), different amounts of dihydrogen ($H_2$) are produced per mole of $N_2$ reduced (see the equations below)[17].

Mo nitrogenase:

$$N_2 + 10\,H^+ + 20\,MgATP + 10\,e^-$$
$$\rightarrow 2\,NH_3 + 2\,H_2 + 20\,MgADP + 20\,P_i$$

V nitrogenase:

$$N_2 + 18\,H^+ + 36\,MgATP + 18\,e^-$$
$$\rightarrow 2\,NH_3 + 6\,H_2 + 36\,MgADP + 36\,P_i$$

Fe nitrogenase:

$$N_2 + 20\,H^+ + 40\,MgATP + 20\,e^-$$
$$\rightarrow 2\,NH_3 + 7\,H_2 + 40\,MgADP + 40\,P_i$$

Recently, it was discovered that besides $N_2$, all three nitrogenases can also reduce carbon monoxide (CO) to hydrocarbons. For CO reduction, the V nitrogenase is the most active isoform, primarily forming C–C bonds and releasing $C_1$ to $C_4$ hydrocarbons, mainly ethylene[18,19]. The Fe nitrogenase shows approximately one-third of the CO activity of the V nitrogenase but releases only methane[20]. The Mo nitrogenase converts CO exclusively into $C_2$ to $C_4$ hydrocarbon chains but is ~800-fold less active than the V nitrogenase[21]. This CO-processing activity can also be exploited for the in vivo conversion of industrial exhaust CO to hydrocarbons as demonstrated for *Azotobacter vinelandii* expressing the V nitrogenase[22].

Beyond CO, it was recently shown that the wild-type V nitrogenase also reduces $CO_2$ to CO, ethene and ethane[23]. In contrast, the wild-type Mo nitrogenase reduces $CO_2$ only to CO (ref. 24) and formate[25]. Surprisingly, the Fe nitrogenase shows the highest $CO_2$ reduction activity among the wild-type nitrogenases, converting $CO_2$ to methane and formate[26]. The tremendous activity differences and varying product spectra, particularly for the reduction of CO and $CO_2$ (further reviewed in refs. 27,28), suggest distinct differences among the three nitrogenase isoenzymes, which are not yet fully understood.

To gain molecular insights into the differences among the three nitrogenase isoenzymes, we set out to solve the structure of the entire Fe nitrogenase complex. For this, we expressed, purified and characterized the Fe nitrogenase of the phototroph *Rhodobacter capsulatus* in its native host. Using anaerobic single-particle cryogenic electron microscopy (cryo-EM), we solved the structure of the adenosine diphosphate-aluminum fluoride ($ADP\cdot AlF_3$)-stabilized Fe nitrogenase complex consisting of two reductase components and one catalytic component at a resolution of 2.35 Å. The structure of the Fe nitrogenase reveals the molecular architecture of the FeFeco and suggests three potential roles of the so far uncharacterized G subunit: (1) binding of the reductase component, (2) substrate channeling and (3) FeFeco positioning and stabilization. Furthermore, the structure allows us to compare the Fe nitrogenase complex with previously published Mo nitrogenase complexes[29–31] and the catalytic component of the V nitrogenase[13]. The comparison reveals distinct features of the Fe nitrogenase architecture, which distinguishes it from the Mo nitrogenase and might influence the catalytic mechanism of the alternative nitrogenases.

## Results

### Engineering *R. capsulatus* for nitrogenase expression

We engineered *R. capsulatus* for studying the Fe nitrogenase. The purple non-sulfur bacterium *R. capsulatus* naturally harbors the Mo and Fe nitrogenases, its genome has been fully sequenced[32] and basic molecular biology methods have been established[33]. Using the *sacB* scarless deletion system (see Methods), we engineered *R. capsulatus* B10S (ref. 34)

to enable high-yield recombinant production and purification of the Fe nitrogenase. (1) We deleted the Mo nitrogenase-encoding gene cluster (*nifHDK*) to ensure that only the alternative nitrogenase is expressed. (2) We knocked out the high-affinity molybdenum transporter genes *modABC* (ref. 35). This modification is essential for high expression levels of the Fe nitrogenase, as trace amounts of molybdenum inside the cell repress the transcription of the Fe nitrogenase genes. (3) We deleted a post-translational modification mechanism that inactivates the nitrogenase reductase component through ADP-ribosylation, encoded by *draT* and *draG* (ref. 36). (4) We removed the bacterial capsule by deleting *gtaI*. Previously, this knockout was found to improve the quality of the cell pellet after centrifugation[37], thus rendering this modification particularly useful for large-scale protein purification with *R. capsulatus*. (5) We interrupted the Fe nitrogenase locus *anfHDGK* by introducing a gentamycin resistance cassette, which allows the recombinant production of the affinity-tagged Fe nitrogenase from expression plasmids. For this purpose, we cloned the *anfHDGK* operon from the bacterial genome into a pOGG024-*kanR* vector and fused a $His_6$-tag to the AnfH amino terminus and a Strep-tag II to the carboxy terminus of AnfD. For nitrogenase expression, we used conjugation to transfer the plasmid into the modified *R. capsulatus* strain. All genetic modifications of the final strain were confirmed by next-generation sequencing (Table 1). In summary, we introduced five genetic modifications into *R. capsulatus* (depicted in Extended Data Fig. 1) that render the purple non-sulfur bacterium an ideal platform for the plasmid-based production and characterization of the Fe nitrogenase. Despite recent advances toward expressing complex heterometallic proteins in *Escherichia coli*, it remains exceedingly challenging to heterologously express nitrogenases with completely assembled metalloclusters in *E. coli*. Therefore, we used the natural diazotroph *R. capsulatus*, which is genetically more accessible than *A. vinelandii*. The plasmid-based expression of nitrogenases in *R. capsulatus* complements the chromosomal nitrogenase expression of *A. vinelandii*, which has been the standard in the field so far.

### Purification and characterization of the Fe nitrogenase

Using our *R. capsulatus* expression strain, we purified and biochemically characterized the Fe nitrogenase. As described in the Methods section, we established an anaerobic workflow for the separate purification of the reductase and catalytic components (Fig. 1a). In vitro, the Fe nitrogenase converted $N_2$ to $NH_3$ at a maximal rate of 0.69 nmol × nmol $(Anf(DGK)_2)^{-1} × s^{-1}$, closely matching the previously published value of 0.72 nmol × nmol $(Anf(DGK)_2)^{-1} × s^{-1}$ that was observed with an untagged Fe nitrogenase from *R. capsulatus*[38]. Notably, the rate of $H_2$ formed under $N_2$ is twice as high as the rate of $NH_3$ formation. As expected, these rates were found to follow a hyperbolic trend with increasing product formation proportional to the ratio of reductase to catalytic component (Fig. 1b). In a pure argon (Ar) atmosphere, all electrons are directed toward $H_2$ formation and a maximal rate of 3.44 nmol × nmol $(Anf(DGK)_2)^{-1} × s^{-1}$ was measured, approximately double the $H_2$ formation rate of the $N_2$ atmosphere (Fig. 1b). Metal quantification via inductively coupled plasma optical emission spectroscopy (ICP-OES) suggested full Fe occupancy for the reductase component and ~80% occupancy for the catalytic component (at 32 expected Fe atoms per catalytic component; Extended Data Fig. 2). This result might be caused by a partial cluster occupancy of the catalytic component or could be the result of slight impurities in the $Anf(DGK)_2$ samples (Fig. 1a,c). However, no transition metal other than iron was detected in our protein samples, confirming a pure Fe nitrogenase. Next, we analyzed the complex formation of the Fe nitrogenase in vitro by trapping the ADP-bound reductase component on the $Anf(DGK)_2$ core with $AlF_3$. Following size-exclusion chromatography (SEC), we detected a protein complex of ~360 kDa in size via mass photometry (Fig. 1c, bottom). The measured masses of the individual nitrogenase components were 236 kDa for $Anf(DGK)_2$ and 69 kDa for $AnfH_2$ (Fig. 1c, top and middle), thus indicating an

**Table 1 | Genetic modifications of *R. capsulatus* expression strain**

| Modification | Deleted locus naturally encodes for | Purpose for deletion | Ref. |
| --- | --- | --- | --- |
| Δ*nifHDK* | Molybdenum nitrogenase | Remove the primary nitrogenase | 38 |
| Δ*modABC* | High-affinity molybdenum transporter | Prevent molybdenum import to maximize expression levels of the alternative nitrogenase | 35 |
| Δ*draTG* | ADP-ribosyltransferase/ADP-ribosyl hydrolase system for the post-translational modification of nitrogenase iron proteins | Ensure constitutive activity of the Fe-only nitrogenase | 36 |
| Δ*gtal* | Quorum sensing protein Gtal | Removes the capsule to increase quality of cell pellet after centrifugation | 37 |
| Δ*anfHDGK::gmR* | Fe-only nitrogenase | Prevent genomic expression of wild-type Fe-only nitrogenase to enable recombinant expression of AnfHDGK variants from plasmid DNA | 55 |

Anf(DGK)$_2$(H$_2$)$_2$ stoichiometry of the complex (expected mass, 374 kDa). These results agree with analytical SEC (Fig. 2c and Extended Data Fig. 2). Next, we analyzed the high-molecular-weight complex by cryo-EM (Table 2). Following anaerobic sample preparation including plunge freezing inside an anaerobic tent, we obtained a 2.35-Å resolution map visualizing the expected heterodecameric complex of two AnfH$_2$ dimers bound to Anf(DGK)$_2$ (Fig. 1d). Taken together, we purified a fully active Fe nitrogenase from *R. capsulatus*, analyzed its activity and complex formation in vitro and solved the structure of the ADP·AlF$_3$-stabilized, AnfH$_2$-bound complex.

**Structure of the Fe nitrogenase**

Using the cryo-EM map, we created a model of the Fe nitrogenase complex and analyzed its molecular features. We used AlphaFold 2 (ref. 39) models for the catalytic component (Anf(DGK)$_2$) and the previously published crystal structure of the reductase component (AnfH$_2$) from *A. vinelandii* (PDB 7QQA; ref. 40) to build a detailed model of the Fe nitrogenase into the electron density map (Fig. 2a). All nitrogenase cofactors were well resolved with local resolutions of up to 1.83 Å (Fig. 2b,d–f and Extended Data Fig. 3). To facilitate the electron transport between the [Fe$_4$S$_4$] cluster of the reductase component to the P-cluster of the catalytic component, two molecules of MgATP must bind to AnfH$_2$ forming a nucleotide-dependent Fe nitrogenase complex. Indeed, our structure contains one ATP mimic MgADP·AlF$_3$ per AnfH subunit, locking the Fe nitrogenase complex in the transition state (Fig. 2a,b)[30,31,41]. Although ATP also fits into the electron density of the cofactor, we decided to model AlF$_3$ at the terminal end of the phosphate esters based on our observation that the AnfH$_2$-bound complex only eluted in the presence of ATP and AlF$_3$ during SEC (Fig. 2c). In the dimeric interface of the reductase component, we observed an [Fe$_4$S$_4$] cluster coordinated by Cys97 and Cys132 of the two interacting AnfH subunits (Fig. 2a,d).

Following complex formation, an electron is transferred from the [Fe$_4$S$_4$] cluster to the P-cluster, an [Fe$_8$S$_7$] cluster embedded at the AnfD–AnfK interface (Fig. 2a,e). In our structure, the [Fe$_4$S$_4$] cluster is 17.7 Å apart from the P-cluster. The P-cluster is in the dithionite-reduced P$^N$ state[42,43] forming a symmetric molecule, connected by a shared sulfide ion and bound by six cysteine residues of either the D or K subunit. During catalysis, the P-cluster donates electrons to the FeFeco at a distance

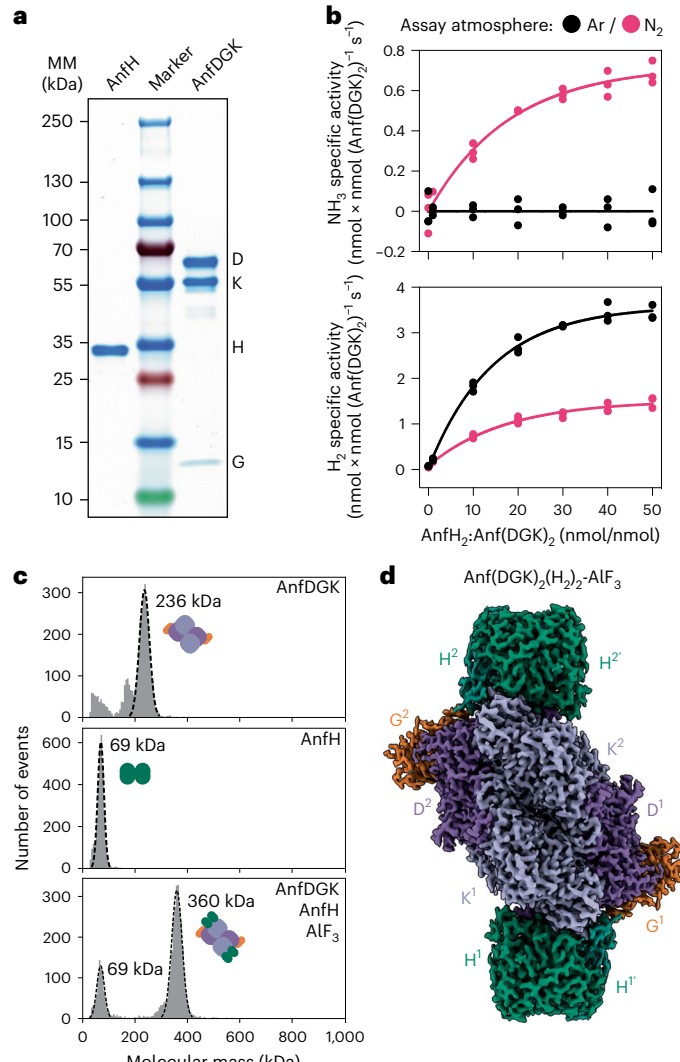

**Fig. 1 | Purification and biochemical characterization of the Fe nitrogenase. a**, SDS–PAGE analysis of the purified Fe nitrogenase reductase component (AnfH) and catalytic component (AnfDGK). MM, molecular mass. **b**, In vitro activity assays of the purified Fe nitrogenase under Ar or N$_2$ atmosphere. Plotted are the specific activities for NH$_3$ (top) or H$_2$ (bottom) formation under varying molar ratios of AnfH$_2$ to Anf(DGK)$_2$. Individual measurements (*n* = 3) are shown, and the solid line represents the non-linear fit of the data. **c**, Mass photometry analysis of the individual nitrogenase components (top and middle) and the AlF$_3$-trapped Anf(DGK)$_2$(H$_2$)$_2$ complex (bottom). Plotted are the number of events versus the molecular mass of the individual events (in kDa). **d**, Electron density map of the AlF$_3$-trapped Fe nitrogenase complex at a global resolution of 2.35 Å (EMD-16890).

of 19.4 Å. As previously proposed[12,17,44], the FeFeco is an [Fe$_8$S$_9$C-(*R*)-homocitrate] cluster (Fig. 2a,f) that, in contrast to the Mo and V nitrogenases, contains no transition metal other than iron (based on ICP-OES). Six irons (Fe$^2$–Fe$^7$) form a trigonal prism around a central carbide that was recently confirmed by X-ray emission spectroscopy[45]. Fe$^1$ and Fe$^8$ anchor the FeFeco to the AnfD backbone via Cys257$^{AnfD}$ and His423$^{AnfD}$, respectively. The latter iron is additionally coordinated by a bidentate (*R*)-homocitrate ligand that binds the iron atom via its 2-hydroxyl and 2-carboxyl moieties, both with distances of 2.2 Å. Taken together, the FeFeco appears to be almost identical to the FeMoco of the Mo nitrogenase, except for Mo being replaced by another Fe.

In summary, we present the comprehensive structure of an alternative nitrogenase complex including the reductase and catalytic components. The structure contains two [Fe$_4$S$_4$] clusters, two P-clusters

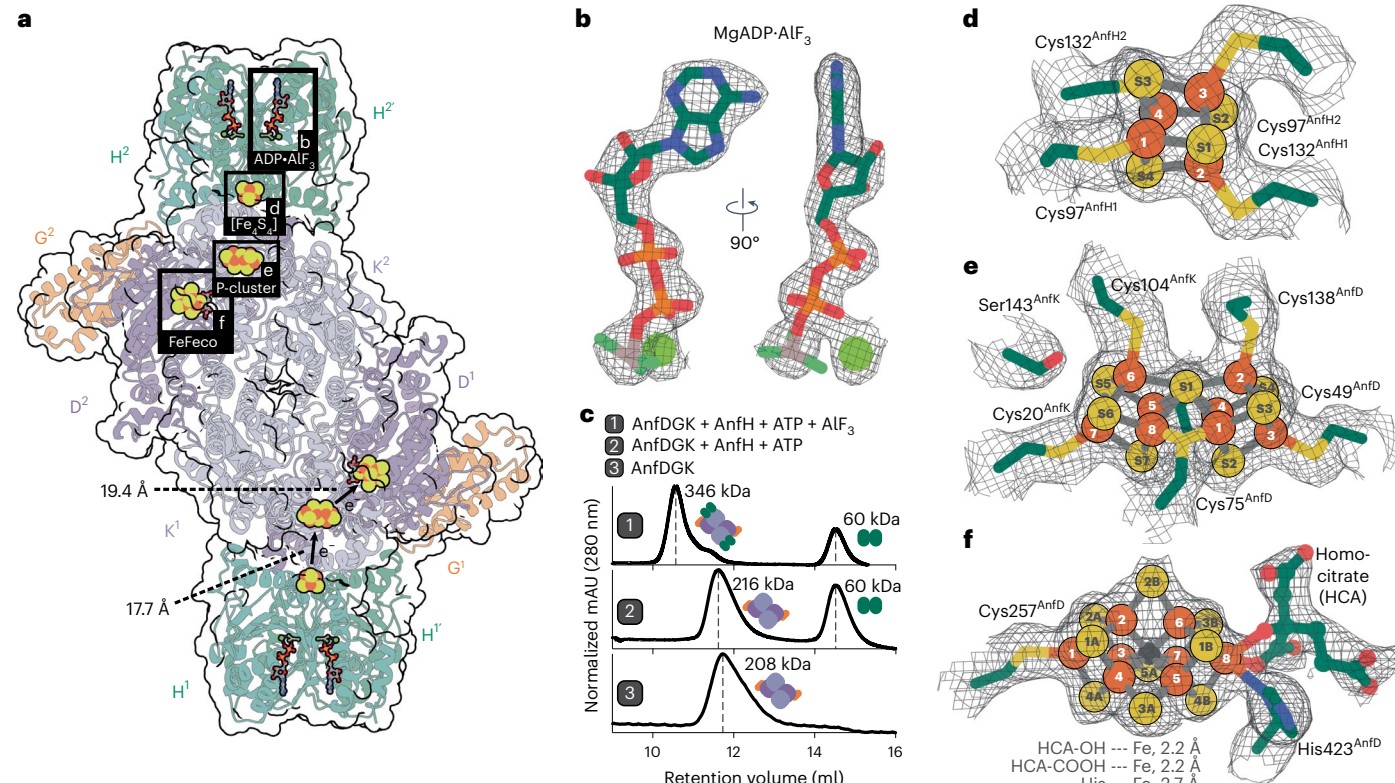

**Fig. 2 | Structure of the Fe nitrogenase and its cofactors. a**, Model of the Fe nitrogenase fitted into the cryo-EM electron density map (PDB 8OIE; EMD-16890). Reductase components (H[1] + H[1'] and H[2] + H[2']) are in green, and the catalytic components are in purple (D[1] and D[2]), light purple (K[1] and K[2]) and orange (G[1] and G[2]). Cofactors are highlighted by gray boxes lettered b, d, e and f. Distances from the [Fe$_4$S$_4$] cluster to the P-cluster and from the P-cluster to FeFeco are indicated. **b**, MgADP·AlF$_3$ cofactor bound to AnfH. Carbon is in green, nitrogen in blue, oxygen in red, phosphorus in orange, aluminum in gray and fluoride in light green. The magnesium ion is depicted as a light-green sphere. **c**, Comparison between the elution profiles of SEC performed with (1) AnfDGK and AnfH in the presence of ATP and AlF$_3$, (2) AnfDGK and AnfH in the presence

of ATP and (3) AnfDGK alone. The reductase component bound Fe nitrogenase complex only elutes in the presence of ATP and AlF$_3$, explaining our decision to model AlF$_3$ instead of a γ-phosphate as shown in **b**. **d**–**f**, Close-up views of the [Fe$_4$S$_4$] cluster (**d**), the P-cluster (**e**) and the FeFeco (**f**). Sulfur atoms of the clusters are represented as yellow spheres, iron atoms as orange spheres and the central carbide ion of the FeFeco as a black sphere. Amino acid residues in the direct cofactor environment are depicted as sticks, with carbon in green, sulfur in yellow, nitrogen in blue and oxygen in red. The homocitrate coordinated to the FeFeco is depicted as a ball-and-stick model with the same color coding as the amino acid residues.

and two FeFecos and thus finally provides direct evidence for the long-hypothesized architecture of the Fe nitrogenase.

## Revealing the roles of the G subunit

The G subunit is a distinct feature of alternative nitrogenases, but its function remains elusive. Therefore, we analyzed our structure for potential roles of the G subunit in the Fe nitrogenase complex. First, the G subunit has been proposed to contribute to the specificity of the interaction between the nitrogenase reductase and catalytic components[46]. This interaction is thought to rely mostly on electrostatic interactions[47]. Indeed, when analyzing the electrostatic potentials of the AnfH$_2$–Anf(DGK)$_2$ interface, we identified complementary surface charges between the two partner proteins (Fig. 3a). This charge pattern is distinct from the Mo nitrogenase, in which the negatively charged regions distal to the [Fe$_4$S$_4$] cluster are much more pronounced and the positively charged patch around the reductase cofactor is less accentuated (Fig. 3b). Intriguingly, the G subunit contributes directly to the binding of AnfH$_2$ through hydrogen bonding between His114[AnfG] and Asn112[AnfH] as well as Thr111[AnfG] and Arg69[AnfH] (Fig. 3c). Instead of the positively charged Arg69[AnfH], NifH features a negatively charged aspartate residue at the same position, underlining the inverse interaction characteristics of the two reductase components. Hence, AnfG likely determines the specificity between Anf(DGK)$_2$ and AnfH$_2$ through hydrogen bonding. Furthermore, the G subunit might be involved in

stabilizing the interaction between the reductase and catalytic components. For the Fe nitrogenase, we only observed ADP·AlF$_3$-trapped complexes consisting of two reductase components and one catalytic component during SEC analysis (see Fig. 2c). This contrasts with the Mo nitrogenase, which forms in the presence of beryllium fluoride mainly complexes consisting of one reductase component and one catalytic component[29].

Second, AnfG might be involved in directing substrates toward the nitrogenase active site. Recent molecular dynamics calculations have suggested an N$_2$ channel to the Mo nitrogenase active site through the D subunit[48], which is conserved in the Fe nitrogenase (Fig. 3d). Using the program CAVER (ref. 49), we identified potential substrate channels to the proposed FeFeco substrate binding site at the sulfur S2B (ref. 7). Intriguingly, both the two likeliest CAVER predictions and the channel calculated using molecular dynamics initialize around the AnfG–AnfH interface, the latter even comprising interactions with residues Ser25[AnfG] and Glu109[AnfG] (Fig. 3d). These interactions could modulate the channel while leaving enough space for small molecules to enter, thus supporting the idea of a regulatory function of the G subunit in substrate accessibility.

Third, our data support an involvement of the G subunit in stabilizing the FeFeco. AnfG is located above the previously described αIII domain, which in the Fe nitrogenase is composed of Arg16[AnfD] to Lys34[AnfD] and Tyr359[AnfD] to Asp384[AnfD] (Extended Data Fig. 4). The αIII

**Table 2 | Cryo-EM data collection, refinement, and validation statistics**

| | AnfHDGK (EMD-16890), (PDB 8OIE) | AnfDK (EMD-17583), (PDB 8PBB) |
|---|---|---|
| **Data collection and processing** | | |
| Magnification | 105,000 | 105,000 |
| Voltage (kV) | 300 | 300 |
| Electron exposure (e–/Å$^2$) | 50 | 52 |
| Defocus range (μm) | −1.4 to −2.4 | −1.4 to −2.4 |
| Pixel size (Å) | 0.837 | 0.837 |
| Symmetry imposed | $C_2$ | $C_2$ |
| Initial particle images (no.) | 3,014,316 | 7,962,489 |
| Final particle images (no.) | 218,653 | 563,245 |
| Map resolution (Å) | 2.35 | 2.49 |
| FSC threshold | 0.143 | 0.143 |
| Map resolution range (Å) | 1.83–7.30 | 2.21–8.00 |
| **Refinement** | | |
| Initial model used (PDB code) | 7QAA, AlphaFold[39] | AlphaFold[39] |
| Model resolution (Å) | 2.34 | 2.48 |
| FSC threshold | 0.143 | 0.143 |
| Model resolution range (Å) | 7.30–2.34 | 8.00–2.48 |
| Map sharpening $B$ factor (Å$^2$) | −50.0 | −107.0 |
| **Model composition** | | |
| Non-hydrogen atoms | 26,151 | 13,334 |
| Protein residues | 3,262 | 1,648 |
| Ligands | 4 × ADP·AlF$_3$, 4 × Mg$^{2+}$, 2 × [Fe$_8$S$_9$C], 2 × (R)-homocitrate, 2 × [Fe$_8$S$_7$], 2 × [Fe$_4$S$_4$] | 2 × partial [Fe$_8$S$_7$] |
| **$B$ factors (Å$^2$)** | | |
| Protein | 61.09 | 29.04 |
| Ligand | 66.10 | 59.13 |
| **R.m.s. deviations** | | |
| Bond lengths (Å) | 0.012 | 0.006 |
| Bond angles (°) | 0.845 | 0.725 |
| **Validation** | | |
| MolProbity score | 1.42 | 1.57 |
| Clashscore | 5.12 | 7.38 |
| Poor rotamers (%) | 0.44 | 0.63 |
| Ramachandran plot | | |
| Favored (%) | 97.16 | 97.09 |
| Allowed (%) | 2.84 | 2.91 |
| Disallowed (%) | 0.00 | 0.00 |

domain forms a lid on top of the active site cofactor that has been shown to undergo major rearrangements during FeMoco insertion[50]. Furthermore, αIII mobility was proposed to play a role in nitrogenase catalysis[29]. Indeed, $B$ factors around the αIII domain are the highest within the catalytic core of the Fe nitrogenase (Fig. 3e), hinting at an inherently flexible character of the αIII domain that is stabilized by the interaction with AnfG. To examine if the αIII domain flexibility is observed or even amplified in the resting state of the catalytic component, we tried to

solve the cryo-EM structure of Anf(DGK)$_2$. Using identical conditions as for the AlF$_3$-trapped complex, particle orientation had a strong bias for top views on AnfD (Extended Data Fig. 5). Hence, AnfD seems to interact with the air–water interface, leading to a preferred orientation of the particles. This issue does not occur in our Anf(DGK)$_2$(H$_2$)$_2$ data set, possibly because the bound reductase component shields the Anf(DGK)$_2$ air–water interface. To circumvent the preferred orientation problem, we collected another data set of the catalytic component with CHAPSO detergent added right before plunge freezing of the grids. As described previously[51], the use of detergent mitigated the preferred orientation problem, and we obtained cryo-EM map with a global resolution of 2.49 Å (Fig. 3f and Extended Data Fig. 5). However, the map is missing electron density for AnfG, suggesting that it was solubilized from the complex after the addition of CHAPSO. Intriguingly, we were not able to resolve electron densities for parts of AnfD (25% of the residues) and AnfK (4% of the residues) in the AnfG-free complex, including the αIII domain and the FeFeco and parts of the P-cluster and its binding site (PDB 8PBB). Nevertheless, the G subunit appears to support FeFeco stabilization through interactions with the αIII domain and might cover the FeFeco insertion site after cluster insertion.

Taken together, we propose three roles for the previously uncharacterized G subunit in the Fe nitrogenase complex: (1) reductase component binding, (2) substrate channeling and (3) FeFeco insertion and stabilization.

## Structural comparison of the nitrogenases

Next, we compared the Fe nitrogenase structure to those of the V and Mo nitrogenases. At first glance, the nitrogenase architectures appear to be quite similar with root mean square deviations between individual subunits of less than 3.2 Å for all isoforms and less than 1.4 Å for the two alternative nitrogenases (Extended Data Table 3). However, sequence alignments of the nitrogenase subunits reveal substantial differences in the N-terminal and C-terminal regions of the D and K proteins (Fig. 4a,b). AnfD features an extended C terminus of approximately 53 amino acids, whereas NifK contains an extended N terminus of around 50 amino acids and a short seven-amino-acid insertion in the C-terminal region relative to the other two homologs, respectively. These differences can be observed over a wide range of species, hinting at a functional relevance of the described features (Extended Data Fig. 6). Intriguingly, all of the described features are located at the dimeric interface of DK–DK (Fig. 4b), raising the question of whether they could influence the proposed cooperative mechanism between the two halves of the nitrogenase complex[7,29]. In the Mo nitrogenase, the N-terminal NifK extension wraps around the neighboring NifD subunit, thereby stabilizing the heterodimer (Fig. 4c). In contrast, the C-terminal extension of AnfD does not touch the neighboring AnfK subunit but forms three α-helices that are positioned at the AnfDK–AnfDK interface. Similarly, the VnfD C terminus is located at the VnfDK–VnfDK interface. However, it is much shorter than the AnfD C terminus, does not form any secondary structure elements and is more similar to the unstructured C terminus of NifD (ref. 15). Overlaying our structure with the Mo nitrogenase complex (PDB 7UTA) and the catalytic component of the V nitrogenase (PDB 5N6Y), we noticed that the alternative nitrogenases align well with each other, whereas only one half of the Mo nitrogenase aligns to the Fe and V nitrogenases (Fig. 4d). In the other half, the complexes appear to be kinked relative to each other, with distances between the respective cofactors of up to 20 Å. We identified two structural differences in the DK–DK interfaces of the three nitrogenases that might cause this effect. On the one hand, the C-terminal regions of the alternative nitrogenases, particularly the extended AnfD C terminus, wedge themselves into the DK–DK interface (Fig. 4e). Here, they interact with neighboring α-helices of the respective K subunits, leading to a downward shift of the homologous helices in the Mo nitrogenase. On the other hand, the seven-amino-acid insertion in the NifK C-terminal region (Ile467$^{NifK}$– Ile473$^{NifK}$) constitutes an extension of the associated α-helix, which

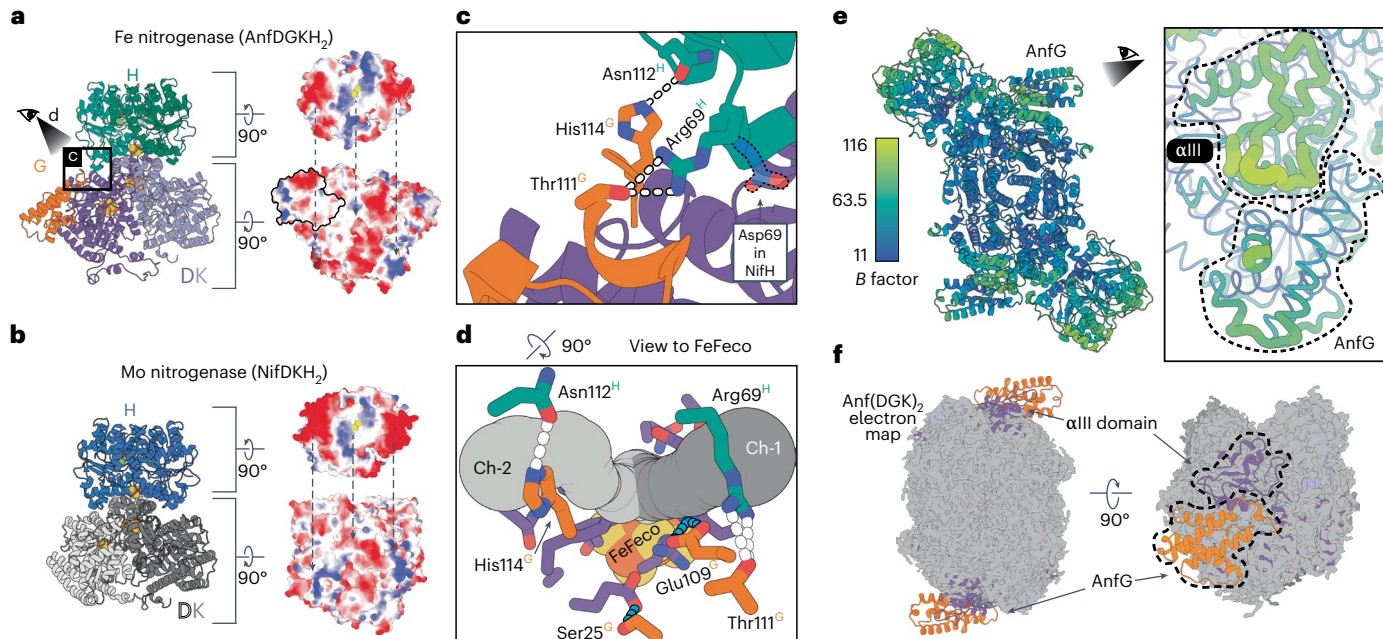

**Fig. 3 | Potential roles of AnfG in the Fe nitrogenase complex. a**, Left: Depiction of the AnfDGKH$_2$ subcomplex. Right: Electrostatic potentials of the AnfH$_2$ (top) and AnfDGK (bottom) interaction surfaces. Negative charges are in red, neutral in white and positive in blue. Arrows indicate interaction interfaces with complementary charges. AnfG is outlined in black. **b**, Left: Depiction of the NifDKH$_2$ subcomplex (modified from PDB 7UTA). Right: Electrostatic surface potentials as shown in **a**. **c**, Close-up view of the AnfG–AnfH interaction interface as highlighted in **a**. Hydrogen bonds between amino acid residues are indicated by white dashes. Shown in faint blue is Asp69$^{NifH}$ that replaces Arg69$^{AnfH}$ in the Mo nitrogenase complex (PDB 7UTA), exemplifying the reverse interaction characteristics of the two reductase components. **d**, A 90° rotation from the view in **c** toward the FeFeco showing two CAVER predicted channels to the active site (Ch-1 and Ch-2; tubes in different shades of gray) and an N$_2$ channel calculated using molecular dynamics proposed by ref. 48 (purple residues). Light-blue dashes highlight interactions of AnfG residues Ser25$^{AnfG}$ and Glu109$^{AnfG}$ with residues of the channel calculated using molecular dynamics. White dashes depict the same interactions highlighted in **c**. **e**, Left: Per-atom $B$ factors within the Fe nitrogenase complex. Right: Highlight of $B$ factors in the αIII domain and AnfG in putty representation. **f**, Model of apo-Anf(DGK)$_2$ fitted into the 2.49-Å cryo-EM map of the CHAPSO detergent-treated Anf(DGK)$_2$ sample (PDB 8PBB; EMD-17583). As highlighted by the arrows, electron density for AnfG and parts of AnfD is missing, including the αIII domain and the FeFeco.

pushes an adjacent helix upward (Fig. 4f). In summary, we propose a complementary effect of the VnfD and AnfD C termini pressing down and the short NifK insertion pushing up structural elements at the DK–DK interface that cause a kink between the two DK halves. (Fig. 4g). Thus, the three nitrogenases differ not only in cofactor composition, but also show distinct structural features, which may contribute to the unique reactivities observed for the individual isoenzymes.

## Discussion

Here, we established *R. capsulatus* as a model organism for the plasmid-based expression and purification of the Fe nitrogenase. After confirming the full N$_2$-reduction activity of the purified enzyme, we solved the Fe nitrogenase structure using cryo-EM. Due to the oxygen sensitivity of the metalloclusters, preparation of nitrogenase cryo-EM samples had to be performed anaerobically, which we successfully accomplished as demonstrated by the reduced P$^N$ state of the P-cluster (Fig. 2e). The identified structural features of the Fe nitrogenase provide molecular insights into the unique properties of the Fe nitrogenase and highlight general features of the alternative nitrogenases.

One specific feature of the alternative nitrogenases is the presence of an additional α-helical subunit: VnfG or AnfG (the G subunit). Yet, the function of the G subunit is poorly understood. Based on our structure and additional experiments, we suggest three potential roles of AnfG in the Fe nitrogenase complex. (1) We identified direct interactions of C-terminal AnfG residues with AnfH$_2$, indicating that AnfG is involved in mediating the docking process of the reductase component (Fig. 3c). Previous crystallographic studies have classified three different docking geometries (DGs) involved in electron transfer from the reductase

component to the catalytic component (DG1–DG3; 11. 7), leading to the hypothesis that the reductase component moves across the surface of the catalytic component during turnover[52,53]. The ADP·AlF$_3$-trapped complex presented here corresponds to the DG2 state, which depicts the moment around the interprotein electron transfer, with AnfH$_2$ being in the most central position. In DG3, AnfH$_2$ should come even closer to AnfG, which therefore might play a role in energy transduction during nitrogenase catalysis and the release of AnfH$_2$ upon ATP hydrolysis. Moreover, the structure indicates that AnfG might contribute to the specificity of AnfH$_2$ with Anf(DGK)$_2$ (Fig. 3c). This hypothesis is in accordance with previously conducted cross-reactivity assays, in which N$_2$ reduction by the Fe nitrogenase was observed exclusively with AnfH$_2$ but not with the two homologous reductase components of *A. vinelandii*[40]. (2) We outlined three potential substrate channels to the FeFeco, which initialize around the location of the G subunit (Fig. 3d). Therefore, we speculate that AnfG potentially modulates and regulates substrate access to the active site. This could partially explain the observed reactivity differences of nitrogenase isoforms for N$_2$, CO and CO$_2$ reduction. (3) Our data suggest that the G subunit contributes to FeFeco insertion and stabilization. This hypothesis is based on our observation that AnfG is located on top of the αIII domain (Fig. 3e), which is associated with the insertion of the active site cofactor and nitrogenase catalysis[29,50]. We observe that AnfG binds and stabilizes the αIII domain, implying that the G subunit is impacting the processes linked to the αIII domain. In support of this hypothesis, loss of AnfG after the addition of detergent leads to a destabilization of the αIII domain accompanied by the loss of the FeFeco (Fig. 3f and Extended Data Fig. 5). Thus, the G subunit might stabilize the active site cofactor through interaction with the αIII domain.

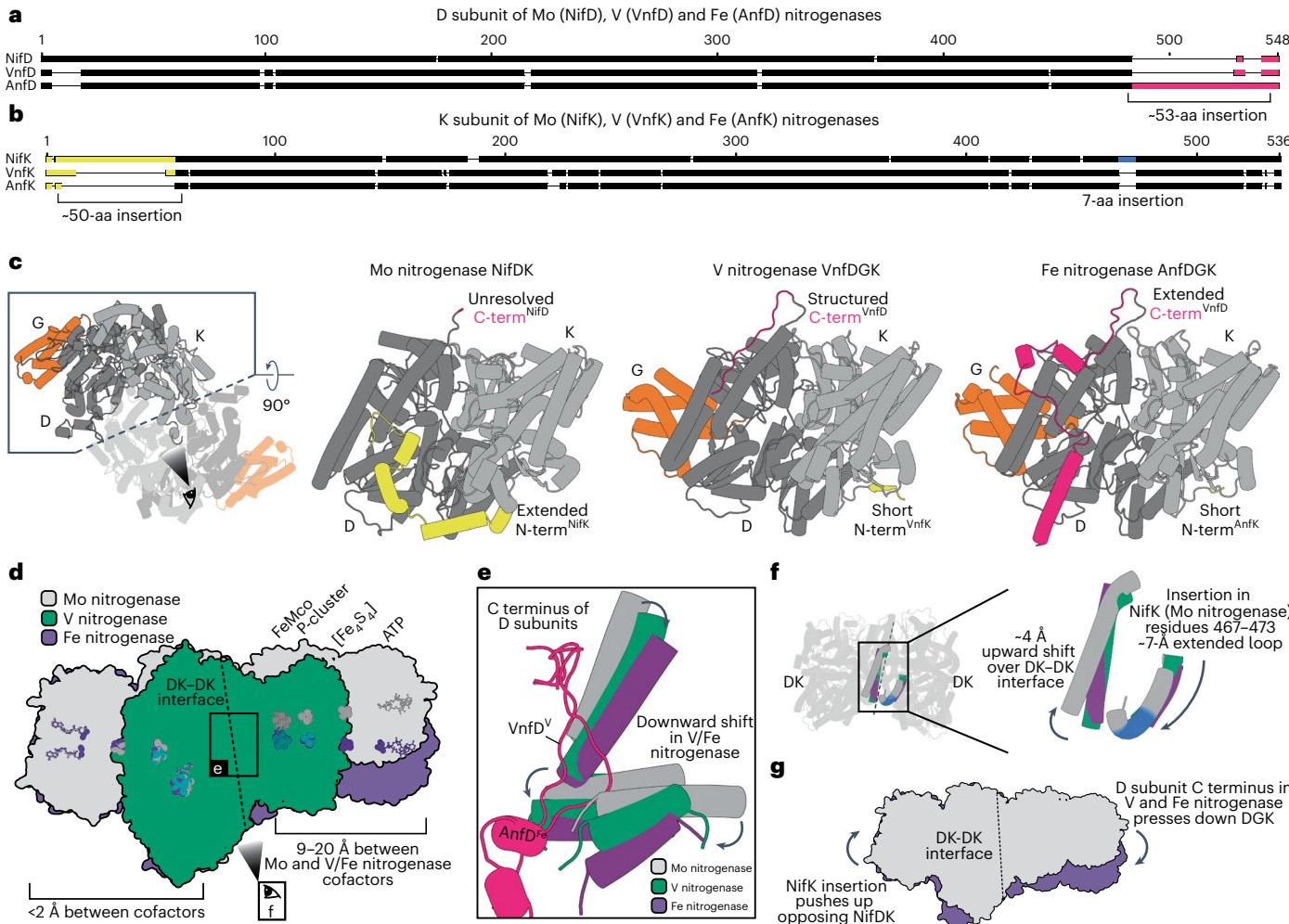

**Fig. 4 | Structural differences between the Mo, V and Fe nitrogenases. a**, MUSCLE alignment among NifD (*A. vinelandii*), VnfD (*A. vinelandii*) and AnfD (*R. capsulatus*) amino acid (aa) sequences. The C-terminal region is highlighted in magenta and is extended by ~53 amino acids in AnfD. **b**, MUSCLE alignment among NifK (*A. vinelandii*), VnfK (*A. vinelandii*) and AnfK (*R. capsulatus*) amino acid sequences. The N-terminal region is highlighted in yellow and is extended by ~50 amino acids in NifK. Another seven-amino-acid insertion in the C-terminal region of NifK is highlighted in blue. **c**, View of the DK–DK interface of the nitrogenase catalytic components, including the D and K subunits in different shades of gray and the G subunits of the alternative nitrogenases in orange. The C and N termini are highlighted in magenta and yellow, respectively, according to **a** and **b**. **d**, Overlay of the Fe nitrogenase (purple; PDB 8OIE; AlF₃-stabilized complex) with the Mo nitrogenase (gray; PDB 7UTA; beryllium fluoride-stabilized complex) and the resting-state catalytic component of the V nitrogenase (green; PDB 5N6Y). The respective cofactors are colored accordingly. The same color coding applies for **e**–**g**. **e**, Close-up view of the DK–DK interface, showing the effects of the respective D subunit C termini (highlighted in magenta) on adjacent α-helices. Overlay is derived from the structures mentioned in **d**. **f**, View of the DK–DK interface, highlighting the effect of the seven-amino-acid insertion in the NifK C-terminal region (blue) on the neighboring α-helix. Overlay is derived from the structures mentioned in **d**. **g**, Summary of the effects associated with the observed kink of the Fe nitrogenase relative to the Mo nitrogenase.

Aligning the Fe nitrogenase complex with the Mo nitrogenase complex, we noticed that the two halves of the catalytic components (NifDK/AnfDGK) are differently interacting with each other, leading to a distortion of the catalytic component and a shift of the cofactors in the second half of the nitrogenase complex (Fig. 4d–g). In a recent cryo-EM study analyzing nitrogenase complexes under turnover conditions[29], it was observed that the two halves of the catalytic component are in different states. Furthermore, only one reductase component was bound at a time, suggesting that the catalytic halves 'communicate' with each other to prevent binding of a second reductase component. Due to the divergent interactions of the catalytic halves described here, we expect a different type of 'ping-pong' mechanism[29] for the alternative nitrogenases potentially also affecting catalytic rate due to a changed half-reactivity[54]. This could potentially have an influence on the different substrate and product profiles observed for the various nitrogenase isoforms for the reduction of N₂, CO and CO₂. A key factor in the communication among the two catalytic halves could be

the 53-amino-acid extended C terminus of AnfD that will be the focus of further investigations. In summary, the structure reported herein offers the foundation to rationally modify and test the differences among nitrogenase isoenzymes to provide insight on the catalytic profiles of the three nitrogenase isoforms.

## Online content

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

## Methods

### Chemicals

Unless noted otherwise, all chemicals were purchased from Carl Roth, Thermo Fisher Scientific, Sigma-Aldrich or Tokyo Chemical Industry and were used directly without further purification. Gases were purchased from Air Liquide.

### Molecular cloning

All used primers were purchased from Eurofins Genomics and are listed in Supplementary Table 1. Polymerase chain reactions (PCRs) were conducted with Q5 High-Fidelity DNA Polymerase (New England Biolabs), PCR purifications were conducted with the Monarch PCR & DNA Cleanup Kit (New England Biolabs), extraction of genomic DNA was conducted with the Monarch Genomic DNA Purification Kit (New England Biolabs), Gibson assemblies were conducted with the NEBuilder HiFi DNA Assembly Master Mix (New England Biolabs), and Golden Gate cloning was conducted with the NEBridge Golden Gate Assembly Kit (New England Biolabs) according to the instructions provided by the manufacturer. Successful assembly of desired vectors was verified by Sanger sequencing through Microsynth Seqlab. All plasmids used and created in this study are listed in Supplementary Table 2.

The pK18mobSacB knockout plasmids were generated via Golden Gate cloning or Gibson assembly. For Gibson assembly, backbone amplification was always done with the primers oMM0227 and oMM0228. The upstream and downstream homologous regions of the targeted genomic loci were amplified from *R. capsulatus* B10S genomic DNA with primers featuring overhangs suitable for Golden Gate cloning or Gibson assembly. The primers used for the construction of each knockout plasmid are listed in Supplementary Table 1. The plasmid pBS85-BsaI-*genR*, used for the interruption of the *anfHDGK* locus by a gentamycin resistance cassette, was constructed via Golden Gate cloning. First, a BsaI cutting site was introduced into pBS85 using the primers oMM0027 and oMM0028 to create pBS85-BsaI. Next, Golden Gate inserts were amplified. The gentamycin resistance cassette was amplified from pOGG024 using oMM0033 together with oMM0034. In parallel, the upstream and downstream homologous regions of the *anfHDGK* locus were amplified from *R. capsulatus* B10S genomic DNA using oMM0035−oMM0038. Eventually, the three inserts and the pBS85-BsaI plasmid were combined in a Golden Gate reaction to generate pBS85-BsaI-*genR*.

For the generation of pOGG024-*kanR*, a kanamycin resistance cassette was amplified via PCR from plasmid pRhon5Hi-2 using primers oMM0384 and oMM385. The plasmid pOGG024 was linearized via PCR with oMM0386 and oMM0387. The two DNA amplicons were purified and combined via Gibson assembly, which yielded pOGG024-*kanR*. The construction of the pOGG024-*kanR-anfHDGK* expression plasmid was achieved in four steps. First, the *anfHDGK* operon was amplified from *R. capsulatus* B10S genomic DNA via PCR using the primers oMM0021 and oMM0146. In parallel, the destination plasmid pRhon5Hi-2 was linearized with oMM0145 and oMM0023. Following purification of the PCR products, pRhon5Hi-2-*anfHDGK* was generated via Gibson assembly. Next, two BsaI cutting sites were removed via a modified version of QuikChange mutagenesis[56] using the primers oMM0161−oMM0164. The resulting plasmid pRhon5Hi-2-*anfHDGK* is suitable for Golden Gate cloning and was used as a template to amplify the *anfHDGK* expression cassette with the primers oMM0389 and oMM0390. The resulting PCR product was purified and subsequently inserted into pOGG024-*kanR* via Golden Gate cloning. Lastly, affinity tags (Strep-tag II and His$_6$-tag) for protein purification were inserted via restriction-free cloning[57] using primers oMM0223 and oMM0224 for the insertion of the Strep-tag II at the AnfD C terminus and oMM0510 and oMM0511 for the insertion of the His$_6$-tag at the AnfH N terminus. Both tags were inserted without any linker sequence. Eventually, the sequence of the pOGG024-*kanR-anfHDGK* expression plasmid was confirmed by whole-plasmid sequencing through Plasmidsaurus.

### Genetic manipulation of *R. capsulatus*

Starting from the wild-type strain B10S, the *R. capsulatus* genome was successively modified to generate an ideal strain for the recombinant expression and subsequent purification of the Fe nitrogenase. For the deletion of *anfHDGK*, a gentamycin resistance cassette was inserted into the *anfHDGK* locus, thereby interrupting the operon (Extended Data Fig. 1). The plasmid pBS85-BsaI-*genR* was introduced into *R. capsulatus* B10S via conjugational transfer as described in ref. 33, selecting for the gentamycin resistance conferred by the transferred vector. Subsequently, double-recombinant clones were identified through screening for gentamycin resistance and tetracycline sensitivity (resulting from the loss of the suicide vector backbone through double recombination) on peptone yeast agar plates[33] containing 15 µg ml$^{-1}$ gentamycin or 10 µg ml$^{-1}$ tetracycline, respectively. Positive clones were further investigated via colony PCR to check the *anfHDGK* locus. The purified PCR products were analyzed by Sanger sequencing to identify clones with a successfully modified *anfHDGK* operon. Building up on the Δ*anfHDGK::genR* mutant of *R. capsulatus* B10S, all further deletions were achieved successively via the *sacB* scarless deletion method described in ref. 58. In brief, sequences of around 500 base pairs homologous to the upstream and downstream regions flanking the gene of interest were generated and cloned into a pK18mobSacB suicide vector (see above). The resulting plasmid was conjugated into the *R. capsulatus* recipient strain[33], selecting for the kanamycin resistance conferred by the suicide vector. Intermediate strains derived from single colonies that were obtained from the previous step were passaged three times in liquid peptone yeast medium[33], growing each passage for 24 h at 30 °C and moderate shaking in the dark. The final passage was spread on a peptone yeast agar plate containing 5% (m/V) sucrose, which was then incubated for 72 h at 30 °C under an Ar atmosphere and illumination by six 60-W krypton lamps (Osram Licht). Single colonies of *R. capsulatus* growing on the sucrose-containing agar plate were screened for kanamycin and sucrose sensitivity on peptone yeast plates containing 50 µg ml$^{-1}$ kanamycin or 5% (m/V) sucrose, respectively. Colonies that could tolerate sucrose but were not growing on kanamycin-containing agar plates were further investigated via colony PCR to check the targeted genomic locus. Lastly, the purified PCR products were analyzed by Sanger sequencing (Microsynth Seqlab) to identify successful knockout clones. Genomic DNA of the modified *R. capsulatus* B10S strain (MM0425) was extracted and sequenced via next-generation sequencing (Novogene) to confirm the deletions listed in Table 1. The *R. capsulatus* MM0436 expression strain was generated by introducing the pOGG024-*kanR-anfHDGK* expression plasmid into MM0425 via conjugational transfer. All used strains are listed in Supplementary Table 3.

### Growth medium and conditions for protein production

*Rhodobacter capsulatus* was cultivated phototrophically at 32 °C under a 100% N$_2$ atmosphere. Cultivation on agar plates was conducted on peptone yeast agar plates[33] selective for the respective expression plasmid. Liquid cultures of *R. capsulatus* were cultivated diazotrophically in a modified version of RCV medium[33] that contained 30 mM DL-malic acid, 0.8 mM MgSO$_4$, 0.7 mM CaCl$_2$, 0.05 mM sodium ethylenediaminetetraacetic acid (Na$_2$EDTA), 0.03 mM thiamine hydrochloric acid, 9.4 mM K$_2$HPO$_4$, 11.6 mM KH$_2$PO$_4$, 5 mM serine, 1 mM iron(III) citrate, 45 µM B(OH)$_3$, 9.5 µM MnSO$_4$, 0.85 µM ZnSO$_4$, 0.15 µM Cu(NO$_3$)$_2$ and 25 µg ml$^{-1}$ kanamycin sulfate at a pH set to 6.8. For protein production, the expression strain was inoculated from a glycerol stock on peptone yeast agar plates and cultivated for 48 h. The obtained cell mass was used to inoculate liquid cultures in N$_2$-flushed RCV medium, which were cultivated for 24 h. Subsequently, 800 ml of RCV medium were inoculated with an optical density at 660 nm (OD$_{660}$) of 0.1 for protein production. Protein purification was initiated when the cultures reached an OD$_{660}$ of ~3.0.

### Protein purification

All protein purification steps were carried out strictly anaerobically under a 95% Ar and 5% H$_2$ atmosphere inside a COY tent (Coy Laboratory

Products). All buffers were anaerobized by flushing them with Ar and equilibrating them for at least 12 h inside the COY tent before use. For collection, sodium dithionite was added to a final concentration of 5 mM to each liquid culture, which were then centrifuged at 15,970 × $g$ for 60 min at 10 °C. The liquid supernatant was decanted, and the cell pellets were resuspended and combined in high-salt buffer (50 mM Tris (pH 7.8), 500 mM NaCl, 10% glycerol, 4 mM sodium dithionite) supplemented with 0.2 mg ml$^{-1}$ bovine pancreatic deoxyribonuclease I and one cOmplete EDTA-free Protease Inhibitor Tablet (Roche). Subsequently, cells were disrupted by three passages through a French Press cell disruptor (Thermo Fisher Scientific, FA-078AE) at 20,000 p.s.i. The obtained lysate was centrifuged at 150,000 × $g$ for 60 min at 8 °C, and the liquid supernatant was filtered (pore size, 0.2 µm). The entire cleared cell extract was then applied sequentially to high-salt buffer equilibrated HisTrap HP (Cytiva) and Strep-Tactin XT 4Flow high-capacity (IBA Lifesciences) columns via an ÄKTA pure chromatography system (Cytiva). After extensive washing with high-salt buffer, the catalytic component was eluted from the Strep-Tactin XT column with high-salt buffer supplemented with 50 mM biotin. Fractions containing the catalytic component were pooled, the buffer was exchanged with low-salt buffer (50 mM Tris (pH 7.8), 150 mM NaCl, 10% glycerol, 4 mM sodium dithionite) with a Sephadex G-25 packed PD-10 desalting column (Cytiva) and the protein was concentrated with an Amicon Ultra-15 Centrifugal Filter Unit (molecular weight cutoff, 100 kDa; Merck Millipore). Meanwhile, the HisTrap HP column was washed extensively with high-salt buffer containing 25 mM imidazole before eluting the reductase component with high-salt buffer plus 250 mM imidazole. The eluate was directly subjected to SEC on a HiLoad 26/600 Superdex 200 pg column (Cytiva) equilibrated with low-salt buffer. AnfH$_2$ eluted in a single peak at around 205 ml and was subsequently concentrated using an Amicon Ultra-15 Centrifugal Filter Unit (molecular weight cutoff, 30 kDa; Merck Millipore). Protein yields for both nitrogenase component fractions were determined using the Quick Start Bradford 1x Dye Reagent (Bio-Rad Laboratories) according to the instructions provided by the manufacturer, and the purity of both protein fractions was analyzed via sodium dodecyl sulfate–polyacrylamide gel electrophoresis (SDS–PAGE). Following the described protein purification procedure, approximately 3.5 mg of catalytic component and 12 mg of reductase component were obtained per liter of *R. capsulatus* growth culture. Eventually, both nitrogenase components were flash frozen and stored in liquid N$_2$ until further use.

## SDS–PAGE analysis
For SDS–PAGE, protein samples were denatured by boiling them with Pierce Lane Marker Reducing Sample Buffer (Thermo Fisher Scientific) for 10 min at 98 °C. After centrifuging the samples at 17,000 × $g$, the clear supernatant was loaded on a 4–20% Mini-PROTEAN TGX Stain-Free Gel (Bio-Rad Laboratories) including PageRuler Plus Prestained Protein Ladder (Thermo Fisher Scientific) as a molecular weight reference. The electrophoresis was run for 30 min at a constant voltage of 180 V before staining the gel with GelCode Blue Safe Protein Stain (Thermo Fisher Scientific).

## Nitrogenase turnover assays
Nitrogenase activity was assessed in vitro by measuring specific activities for H$_2$ and NH$_3$ formation under an N$_2$ or Ar atmosphere. Working under an Ar atmosphere, varying amounts of AnfH$_2$ were dissolved in an anaerobic solution of 50 mM Tris (pH 7.8), 10 mM sodium dithionite, 3.5 mM ATP, 7.87 mM MgCl$_2$, 44.59 mM creatine phosphate and 0.20 mg ml$^{-1}$ creatine phosphokinase (Sigma-Aldrich, C3755). The reaction vials were sealed by crimping them with butyl rubber stoppers, and the headspace was exchanged to N$_2$ or Ar. Next, the reactions were initialized by adding 0.1 mg of Anf(DGK)$_2$ up to a total volume of 700 µl and were allowed to proceed for 8 min at 30 °C with moderate shaking at 250 r.p.m. Reactions were quenched with 300 µl of 400 mM

Na$_2$EDTA solution (pH 8.0), and the amounts of formed H$_2$ and NH$_3$ were analyzed as described below.

## Quantification of H$_2$
Amounts of formed H$_2$ were determined via headspace analysis using a Clarus 690 GC system (gas chromatography with a flame ionization detector and a thermal conductivity detector, GC–FID/TCD; PerkinElmer) with a custom-made column circuit (ARNL6743) that was operated with TotalChrom v6.3.4 software (PerkinElmer). The headspace samples were injected by a TurboMatrixX110 autosampler (PerkinElmer), heating the samples for 15 min to 45 °C prior to injection. The samples were then separated on a HayeSep column (7′ HayeSep N 1/8′ Sf; PerkinElmer), followed by molecular sieve (9′ Molecular Sieve 13×1/8′ Sf; PerkinElmer) kept at 60 °C. Subsequently, the gases were detected with an FID (at 250 °C) and a TCD (at 200 °C). The quantification of H$_2$ was based on a linear standard curve that was derived from measuring varying amounts of H$_2$ under identical conditions. The results were plotted using GraphPad Prism v9 software (Dotmatics).

## Quantification of NH$_3$
Quantification of in vitro generated NH$_3$ was done with a modified version of a fluorescence NH$_3$ quantification method described in ref. 59. One hundred microliters of sample were combined with 1 ml of a solution containing 2 mM *o*-phthalaldehyde, 10% (V/V) ethanol, 0.05% (V/V) β-mercaptoethanol and 0.18 M potassium phosphate buffer (pH 7.3) and incubated for 2 h at 25 °C in the dark. Fifty microliters of each sample were transferred into individual wells of a black Nunc F96 MicroWell plate (Thermo Fisher Scientific), and fluorescence at 485 nm was monitored with an Infinite 200 PRO plate reader (Tecan) in fluorescence top reading mode using an excitation wavelength of 405 nm. The quantification of NH$_3$ was based on a linear standard curve that was derived from measuring varying amounts of NH$_4$Cl under identical conditions. Samples incubated under an Ar atmosphere instead of N$_2$ were used to correct for background signal. The results were plotted using GraphPad Prism v9.

## Mass photometry
Mass photometry measurements were carried out on microscope coverslips (1.5 H, 24 × 50 mm; Carl Roth) with CultureWell Reusable Gaskets (CW-50R-1.0, 50–3 mm diameter × 1 mm depth) that had been washed with three consecutive rinsing steps of distilled H$_2$O and 100% isopropanol and dried under a stream of pressurized air. Measurements were set up in gaskets assembled on microscope coverslips on the stage of a TwoMP mass photometer (Refeyn) with immersion oil. Samples were measured in anaerobic measurement buffer (150 mM NaCl, 50 mM Tris (pH 7.8), 10% glycerol, 10 mM sodium dithionite) after focusing on the glass surface using the droplet-dilution focusing method. After focusing, 0.5 µl of nitrogenase sample (500 nM stock concentration, dissolved in measurement buffer with 4 mM dithionite) were removed from an anaerobic vial, quickly added to 19.5 µl of measurement buffer, and mixed on the stage of the mass photometer. Measurements were started ~5 s after removing protein from the anaerobic environment. Data were acquired for 60 s at 100 frames per second using AcquireMP v2.3 (Refeyn). Mass photometry contrast was calibrated to molecular masses using 50 nM of a mixture of citrate synthase complexes with varying complex stoichiometries of masses ranging from 86 kDa to 430 kDa. Mass photometry data sets were processed and analyzed using DiscoverMP v20222 R1 (Refeyn). The details of mass photometry image analysis have been described previously[60].

## Metal analysis
Metal analysis was done using ICP-OES. For sample preparation, 0.12 mg and 0.24 mg of catalytic and reductase components, respectively, were

dissolved in 0.5 ml of trace metal grade concentrated nitric acid and incubated for 12 h at 25 °C. Subsequently, the samples were boiled for 2 h at 90 °C before they were diluted 17-fold in distilled water. The metal content was analyzed with a 720/725 ICP-OES device (Agilent Technologies) on iron ($\lambda$ = 238.204 nm), molybdenum ($\lambda$ = 202.032 nm), nickel ($\lambda$ = 216.555 nm) and zinc ($\lambda$ = 213.857 nm). The device was operated with ICP Expert v4.1.0 software (Agilent Technologies). All analyzed metals were quantified using ICP multi-element standard solution IV (Merck) as a standard. The results were plotted using GraphPad Prism v9.

#### Preparation of AlF₃-stabilized nitrogenase complex

Stabilized Fe nitrogenase complex consisting of two reductase components and one catalytic component was prepared as described in ref. 61. In brief, 4 nmol of catalytic component and 32 nmol of reductase component were combined in 100 mM MOPS, 50 mM Tris, 100 mM NaCl (pH 7.3) with 5 mM sodium dithionite, 4 mM NaF, 0.2 mM $AlCl_3$, 8 mM $MgCl_2$ and 1 mM ATP in a total volume of 4 ml. The reactions were incubated for 1 h at 30 °C before they were concentrated with an Amicon Ultra-0.5 ml Centrifugal Filter Unit (molecular weight cutoff, 100 kDa; Merck Millipore). Subsequently, less than 500 µl of sample were injected via the ÄKTA pure chromatography system onto a Superdex 30 Increase 10/300 GL column (Cytiva) equilibrated with 50 mM Tris (pH 7.8), 200 mM NaCl and 5 mM sodium dithionite. Elution fractions from the peak corresponding to the appropriate molecular weight species (expected molecular weight of catalytic component combined with two reductase components is ~372 kDa) were pooled, and the presence of all nitrogenase subunits was confirmed via SDS–PAGE as described above.

#### Cryo-EM sample preparation and data collection

Four microliters of protein solution (total protein concentration, 1 mg ml$^{-1}$) were applied to freshly glow-discharged QUANTIFOIL R2/1 300 copper mesh grids (Quantifoil Micro Tools) and blotted for 5 s with a blot force of 5 at ~90% humidity and 8 °C using a Vitrobot Mark IV (Thermo Fisher Scientific) that was placed inside an anaerobic COY tent. For CHAPSO detergent-supplemented grids, 1 µl of detergent (dissolved in the same buffer as the protein) was added to a final concentration of 0.4% (m/V) to 3 µl of protein solution on the respective grid. Grids were plunge frozen in a liquid ethane (37 vol%) and propane (63 vol%) mix and stored in liquid nitrogen until data collection. CHAPSO-supplemented grids of AnfDGK were prepared to prevent preferential orientation.

Data of cryo-EM samples were collected on a Titan Krios G3i electron microscope (Thermo Fisher Scientific), operated at an acceleration voltage of 300 kV and equipped with a BioQuantum K3 energy filter (Gatan). Data were collected in electron counting mode at a nominal magnification of ×105,000 (0.837 Å per pixel) with a total dose of 50 e–/Å$^2$ (50 fractions), using the aberration-free image-shift correction in EPU v2.9–2.11 software (Thermo Fisher Scientific). The nominal defocus range used for data collection was −1.4 µm to −2.4 µm.

#### Cryo-EM data processing

All data sets were processed entirely in cryoSPARC v4.1 (ref. 62). For all data sets, dose-fractionated movies were gain-normalized, aligned, and dose-weighted using Patch Motion correction, and the contrast transfer function (CTF) was determined using the Patch CTF routine. The information regarding cryo-EM data collection, model refinement and statistics are listed in Table 2.

**Processing the AnfHDGK–AlF₃ complex.** Blob picker and manual inspection of particles were used to extract an initial 2,114,475 particles with a box size of 300 pixels, which were used to build two-dimensional (2D) classes. 2D classes with protein-like features were used to initialize template picking. After manual inspection and extraction with a box size of 300 pixels, this yielded a total of

3,365,366 particles, which were used to build 2D classes. After selecting 2D classes with protein-like features, the selected particles were used to train a model that was subsequently used to pick particles using Topaz (ref. 63). A total of 1,706,699 candidate particles were extracted with a box size of 380 pixels and cleaned from non-particle candidates by 2D classification into 200 classes. Selected particles were used for ab initio reconstruction and classification into four classes. Particles of the two best-aligning classes (432,216 particles) were subjected to further cleaning by three-dimensional (3D) classification into ten classes with a target resolution of 5 Å. 3D classification yielded volumes containing zero, one or two AnfG subunits, with unchanged orientation of the remaining subunits. The best-aligning classes with one or more AnfG subunit bound (218,653 particles) were subjected to local CTF refinement, local motion correction, and subsequent non-uniform refinement with $C_2$ symmetry, two extra final passes, 15-Å initial low-pass resolution, 12-Å gold-standard Fourier shell correlation (GSFSC) split resolution, 4-Å dynamic mask near expansion, 10-Å dynamic mask far expansion, 8-Å dynamic mask start resolution, per-particle defocus optimization, and Ewald sphere (EWS) correction, yielding a global resolution of 2.35 Å and a temperature factor of −76.7 Å$^2$. Further classification did not yield improved resolution.

**Processing the AnfDGK component.** Initial attempts to solve the AnfDGK complex structure without reductase component used grids prepared without detergent (CHAPSO). Standard processing workflows of this data set (blob picking, template picking, Topaz picking and manual picking) yielded 2D classes that exclusively showed one orientation (Extended Data Fig. 5a). Resulting ab initio and 3D reconstructions failed to yield initial volumes with a nitrogenase-like shape. We therefore focused our efforts on grids prepared in the presence of 0.5% CHAPSO.

Here, blob picker and manual inspection of particles were used to extract an initial 2,018,560 particles with a box size of 320 pixels from 2,000 micrographs, which were used to build 2D classes. 2D classes with protein-like features were used to train a Topaz model to pick particles, which was subsequently used to re-extract particles from the same 2,000 micrographs for downstream 2D classification and Topaz model training. A total of 1,647,264 particles were extracted with a box size of 340 pixles and cleaned from non-particle candidates by 2D classification. Cleaned particles were used to train a Topaz model on 4,578 micrographs and subsequently used to pick particles from all 18,320 micrographs. A total of 7,962,489 particles were extracted with a box size of 324 pixels and cleaned by three subsequent rounds of 2D classification into 200, 100 and 50 classes, respectively (Extended Data Fig. 5b). Selected particles of the last 2D classification step (2,121,950) were used for ab initio reconstruction and classification into four classes. Particles of the two best-aligning classes (1,336,362 particles) were subjected to further cleaning by 3D classification into ten 3D classes with a target resolution of 4 Å. 3D classification yielded no volumes containing electron density at positions where AnfG would be expected. Nevertheless, particles of the three best-aligning classes (304,619 particles) were used for non-uniform refinement with $C_1$ symmetry and no additional corrections. This yielded a 2.64-Å global resolution map that contained no indication of electron density at locations where AnfG would be expected, nor at select regions of AnfDK in close contact with AnfDK. A subsequent non-uniform refinement using particles of the seven best-aligning classes (563,245) from the 3D classification, the 2.64-Å map as an input volume, $C_2$ symmetry, CTF correction, defocus correction and EWS correction yielded a map with a global resolution of 2.49 Å. This map also contained no electron density at locations where AnfG would be expected, nor at regions in AnfDK that would be near the expected AnfG position. Further classification was not attempted, as AnfG could not be detected in processed volumes.

## Model building and refinement

Initial cryo-EM map fitting was performed in UCSF Chimera v1.16 (ref. 64) using AlphaFold 2 (ref. 39) models for AnfD, AnfK and AnfG, as well as an AnfH crystal structure (PDB 7QQA) from *A. vinelandii*[40]. The resulting model was manually built further in Coot v0.8.9.2 (ref. 65). Automatic refinement of the structure was done using phenix.real_space_refine of the Phenix v1.21.1 software suite[66]. Manual refinements and water picking were performed with Coot v0.8.9.2. The FeFeco was built with REEL of the Phenix software suite. The model statistics are listed in Table 2.

## Substrate channel calculation

Substrate channels were calculated using CAVER v3.0.3 software (ref. 49). The coordinates of sulfur atom S2B were provided as the starting point for channel calculations. The probe radius, shell radius and shell depth were set to 0.7, 4.0 and 5.0 Å, respectively. Many channels were predicted using CAVER v3.0.3. However, the two most probable channels with the shortest length, the largest bottleneck radius, the highest throughput and prioritization by CAVER v3.0.3 were selected and are displayed throughout the manuscript as surfaces generated in PyMOL v2.5 (Fig. 3d).

## Reporting summary

Further information on research design is available in the Nature Portfolio Reporting Summary linked to this article.

## Data availability

All unique materials used in this study are available from the corresponding author upon request. All raw data for mass photometry measurements, kinetic experiments, and protein characterization are deposited on Edmond, the Open Research Data Repository of the Max Planck Society for public access (https://doi.org/10.17617/3.P6SEVC)[67]. The structures reported in this paper are deposited to the Protein Data Bank (PDB) under the accession codes 8OIE (ADP·AlF$_3$-stabilized Fe nitrogenase complex) and 8PBB (CHAPSO-treated partial Fe nitrogenase catalytic component). Cryo-EM data were deposited to the Electron Microscopy Data Bank (EMDB) under EMD-16890 (ADP·AlF$_3$-stabilized Fe nitrogenase complex) and EMD-17583 (CHAPSO-treated partial Fe nitrogenase catalytic component). The structures used to build our model and compare our structure are available at the PDB under accession codes 7QQA, 7UTA and 5N6Y. Sequences for protein alignment were obtained from the National Center for Biotechnology Information (NCBI) non-redundant protein database and are available as a table in the source data on Edmond (https://doi.org/10.17617/3.P6SEVC). Source data are provided with this paper.

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

## Acknowledgements

We thank the Central Electron Microscopy Facility at the Max Planck Institute of Biophysics for expertise and access to instruments. We thank G.K.A. Hochberg for access to and support with the mass photometer. We thank S. Freibert and J.M. Schuller for help with anaerobic plunge freezing of cryo-EM sample grids and the use of their equipment. We thank C. Thölken, P. Klemm and M. Lechner for help with data management and computing cluster access. We thank S. Reinhard for help in data and sample transport. We thank M. Girbig, F. Ramirez, A. Kumar and J.M. Schuller for help during cryo-EM data processing. We thank B. Masepohl and T. Drepper for plasmids and *R. capsulatus* strains. This work was supported by the German Research Foundation (grant 446841743 to J.G.R.). F.V.S., L.S., J.Z., S.P., N.N.O., T.J.E. and J.G.R. are grateful for generous support from the Max Planck Society. L.S. thanks the Joachim Herz Foundation for support in the form of an Add-On Fellowship for Interdisciplinary Life Sciences. N.N.O. thanks the Verband der Chemischen Industrie for a Kekulé fellowship. Open access publishing was enabled and organized by Projekt DEAL.

## Author contributions

J.G.R. conceived and supervised the project. T.J.E. and J.G.R. acquired funding. F.V.S., L.S. and J.G.R. designed and analyzed experiments. F.V.S. and N.N.O. performed molecular work. F.V.S. performed anaerobic protein purification and enzyme biochemistry. F.V.S. and L.S. performed mass photometry experiments. F.V.S., L.S. and S.P. performed cryo-EM data acquisition. L.S., F.V.S. and J.Z. processed and refined the cryo-EM structure. L.S., F.V.S., J.Z. and J.G.R. analyzed the cryo-EM structure. F.V.S., L.S. and J.G.R. wrote the original manuscript, which was reviewed and edited by all authors.

## Funding

## Competing interests

The authors declare no competing interests.

## Additional information

**Extended data** is available for this paper at

**Supplementary information** The online version contains supplementary
material available at https://doi.org/10.1038/s41594-023-01124-2.

**Correspondence and requests for materials** should be addressed to
Johannes G. Rebelein.

**Peer review information** *Nature Structural & Molecular Biology*
thanks Tracey Rouault, Hannah Rutledge and the other, anonymous,
reviewer(s) for their contribution to the peer review of this work.
Primary Handling Editor: Katarzyna Ciazynska, in collaboration with
the *Nature Structural & Molecular Biology* team. Peer reviewer reports
are available.

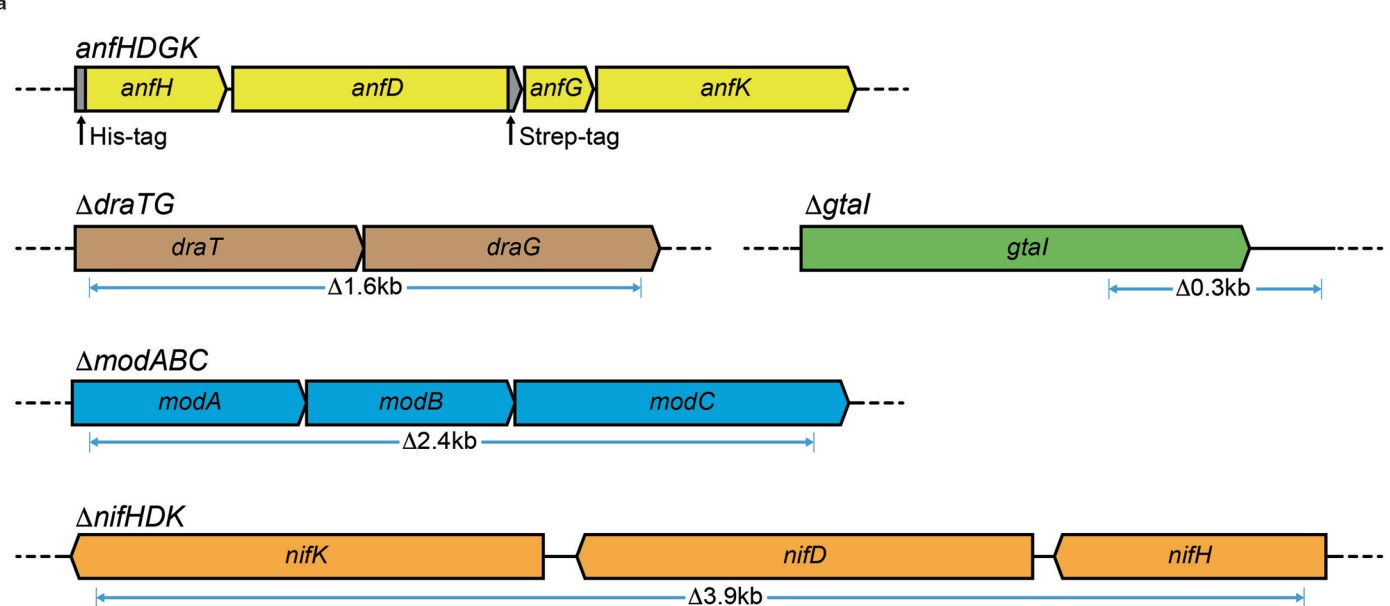

**Extended Data Fig. 1 | Modified genomic regions and the corresponding polymorphism derived NGS data. (a)** Strain MM0425 was sequenced using Illumina sequencing (Novogene Co., Ltd., Beijing, China). The individual reads were trimmed, paired and assembled to the *R. capsulatus* reference genome (Strain SB1003, GenBank CP001312.1) using the Breseq pipeline.

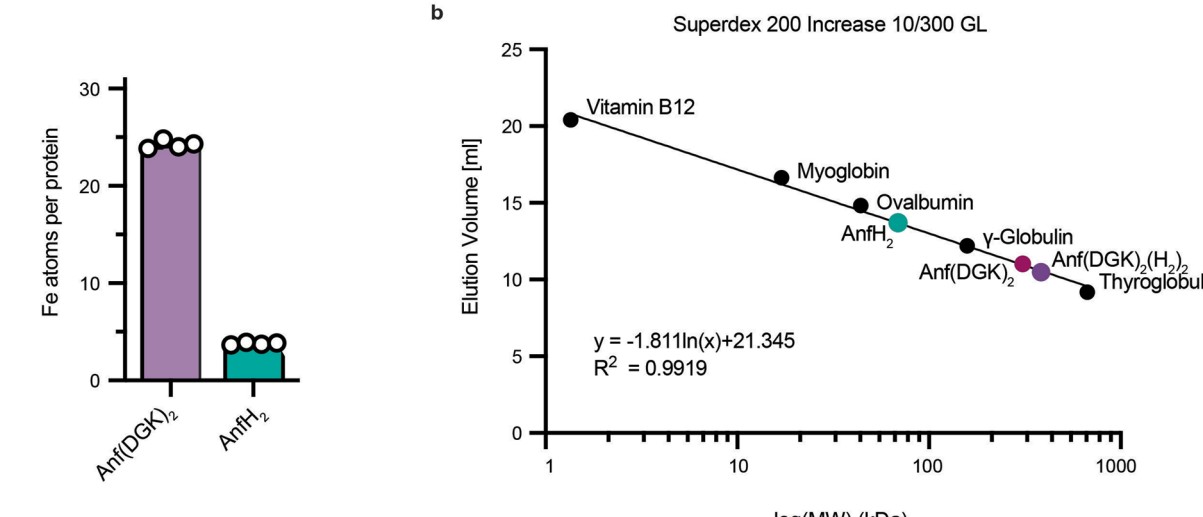

**Extended Data Fig. 2 | Analysis of Anf(DGK)$_2$ and AnfH$_2$. (a)** Inductively coupled plasma optical emission spectroscopy (ICP-OES) data for Anf(DGK)$_2$ and AnfH$_2$. Data are 2 technical replicates of 2 biological replicates. **(b)** Analytical size-exclusion chromatography standard run on a Superdex 200 Increase 10/300 GL column (Cytiva Europe GmbH, Freiburg, Germany) used to infer the complex masses of AnfH$_2$, Anf(DGK)$_2$ and Anf(DGK)$_2$(H$_2$)$_2$. Black dots indicate proteins included in the standard mixture; coloured dots indicate measured complexes.

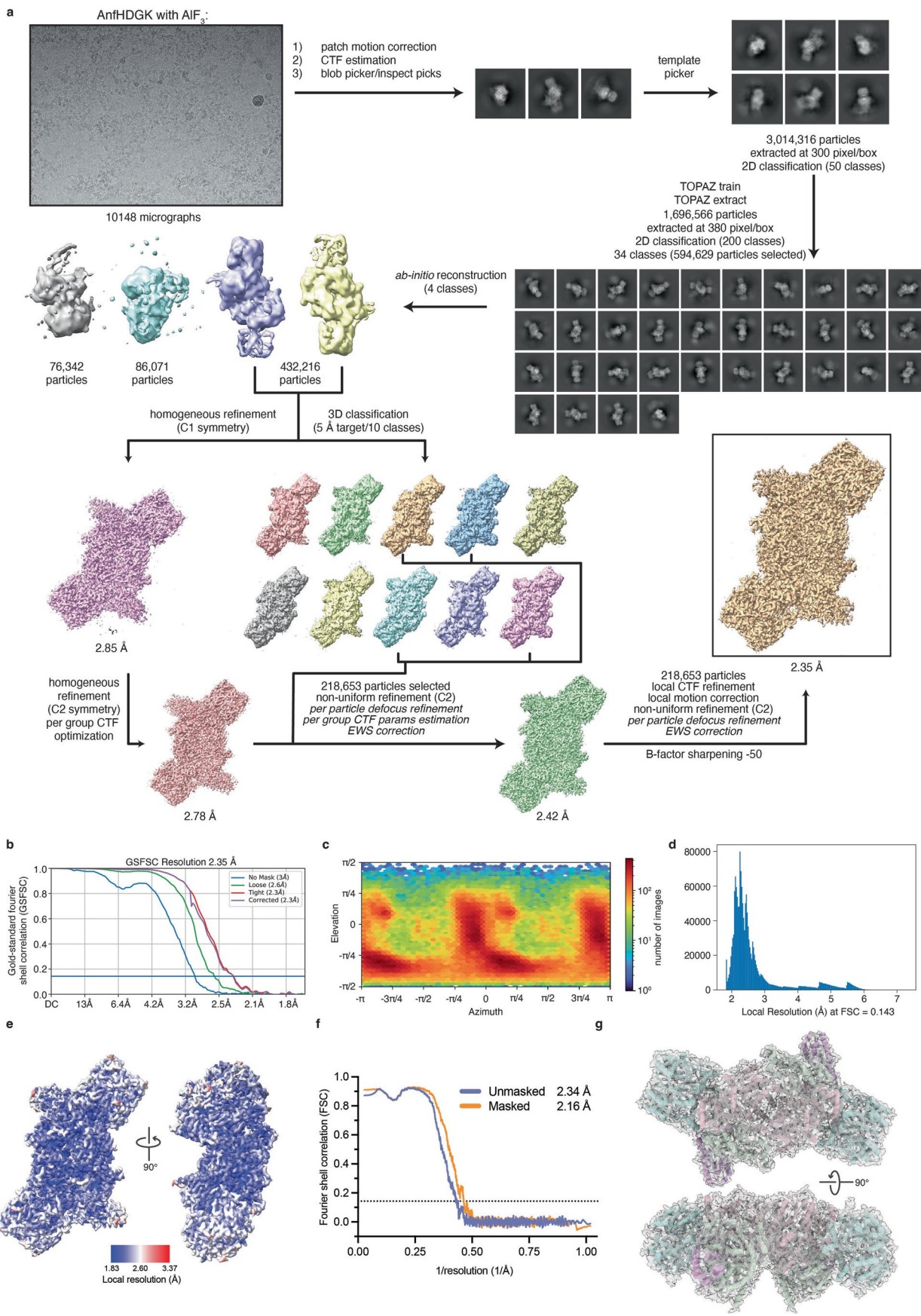

**Extended Data Fig. 3 | See next page for caption.**

**Extended Data Fig. 3 | Cryogenic electron microscopy data collection and analysis of Anf(DGK)$_2$(H$_2$)$_2$. (a)** Schematic data processing workflow for the electron map of Anf(DGK)$_2$(H$_2$)$_2$. Dataset was collected on a Titan Krios G3i electron microscope operated at an acceleration voltage of 300 kV and equipped with a BioQuantum energy filter and a K3 direct electron detector. Dataset was processed entirely in CryoSPARC[62]. **(b)** Gold-standard Fourier shell correlation plot from map refinement in CryoSPARC. Resolution determined at Fourier shell correlation (FSC) = 0.143. **(c)** Angular particle distribution. **(d)** Distribution of local resolution at FSC = 0.143. **(e)** Local resolution as calculated by CryoSPARC mapped onto the refined density with different views shown. **(f)** Map to atomic model FSC plot determined at FSC = 0.143. **(g)** Two viewing angles of model fit to map.

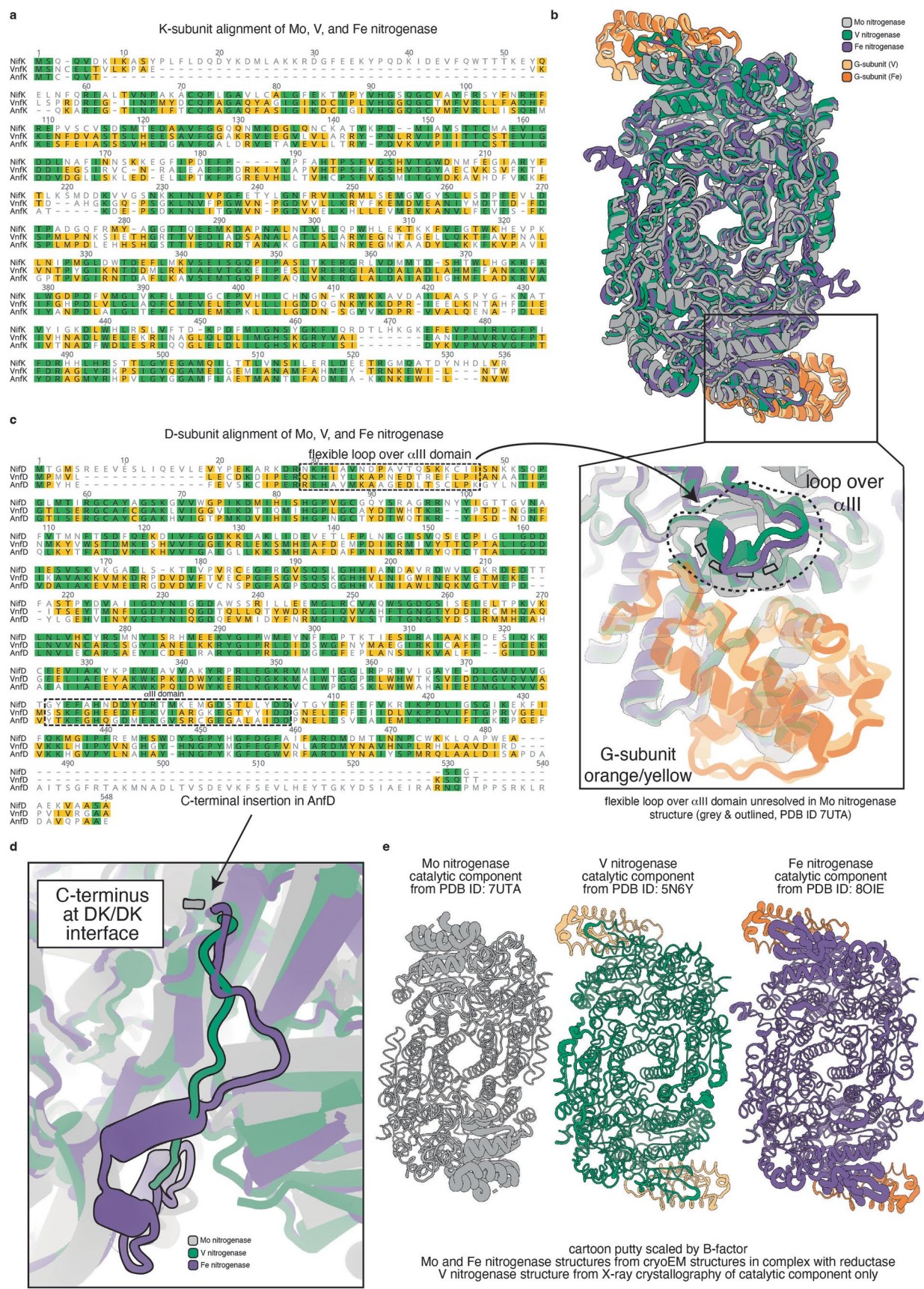

**Extended Data Fig. 4 | See next page for caption.**

**Extended Data Fig. 4 | Sequence and structural alignment of Mo, V, and Fe nitrogenase catalytic components. (a)** Sequence alignment of *Azotobacter vinelandii* Mo and V nitrogenase K-subunit (NifK and VnfK) to *Rhodobacter capsulatus* Fe nitrogenase K-subunit (AnfK). Identical sites are shown in green and similar sites (BLOSUM62 matrix[68] threshold of 2) that occur in 2 of 3 sequences are highlighted in yellow. **(b)** Structural alignment of the catalytic components of the Mo (PDB: 7UTA, resting-state catalytic component), V (PDB: 5N6Y, resting-state catalytic component), and Fe (PDB: 8OIE, AlF3-trapped complex) nitrogenases aligned in (a) and including the G-subunit for V and Fe nitrogenase. Inset at bottom shows a zoomed view of the αIII domain, with arrows depicting the origin in the sequence alignment. **(c)** Sequence alignment of *Azotobacter vinelandii* Mo and V nitrogenase D-subunit (NifD and VnfD) to *Rhodobacter capsulatus* Fe nitrogenase D-subunit (AnfD). Identical sites are shown in green and similar sites (BLOSUM62 matrix, threshold of 2) that occur in 2 of 3 sequences are highlighted in yellow. **(d)** Close-up view into the interface between individual DK-halves of the catalytic component. Arrow points towards C-terminus of D-subunit, which is extended in Fe nitrogenase. **(e)** Structural alignment of putty-styled cartoon Mo, V, and Fe nitrogenases (same models as in (b)). Putty size is scaled by B-factor. Mo (PDB: 7UTA, resting-state catalytic component) and Fe (PDB: 8OIE, AlF$_3$-trapped complex) nitrogenase structures are measured by cryo-EM, whereas V (PDB: 5N6Y, resting-state catalytic component) nitrogenase structure derives from X-ray crystallography.

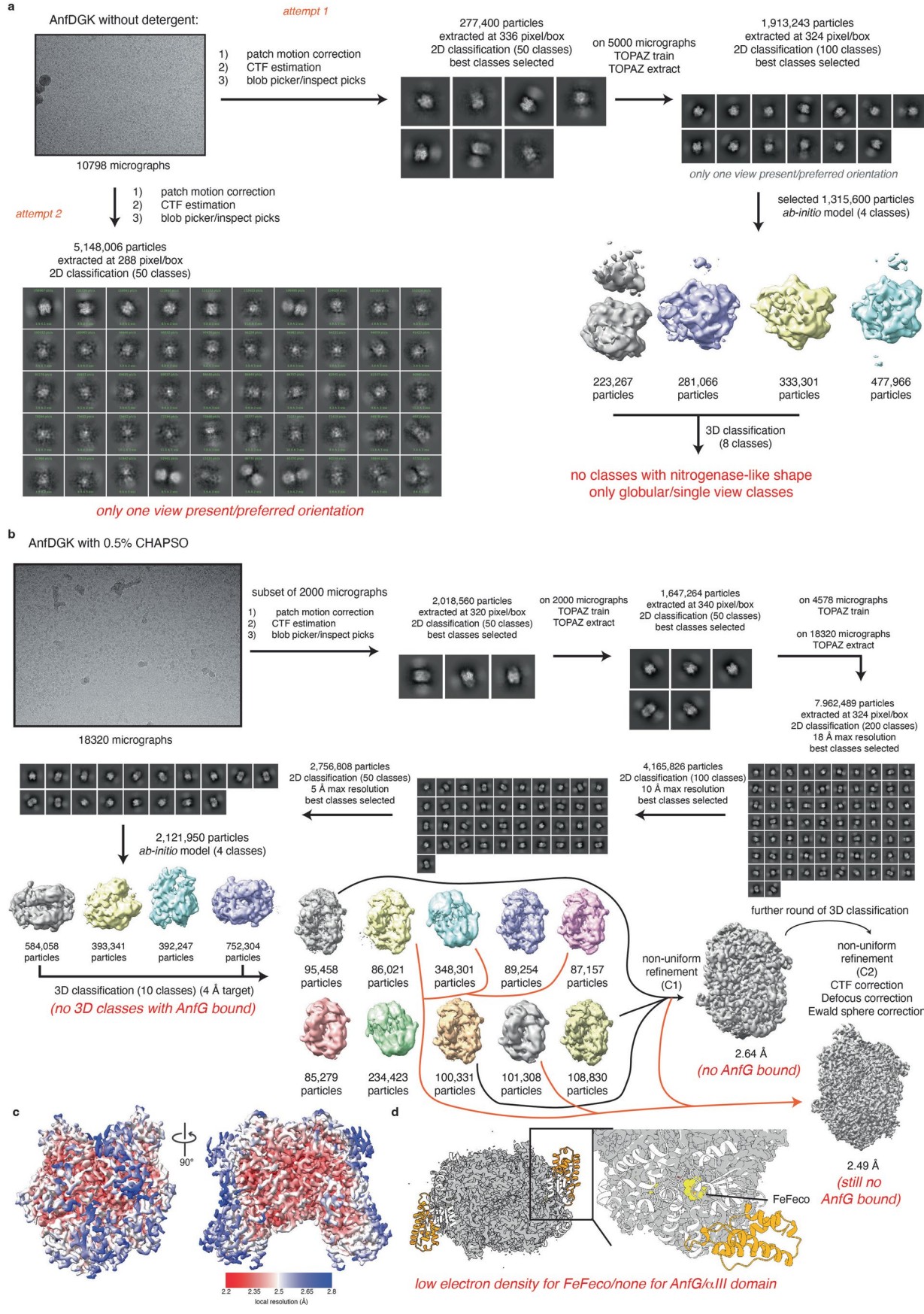

**Extended Data Fig. 5 | See next page for caption.**

**Extended Data Fig. 5 | Cryogenic electron microscopy data collection and analysis of the catalytic component Anf(DGK)$_2$. (a)** Schematic data processing workflow for the electron map of Anf(DGK)$_2$ collected without detergent. Dataset was collected on a Titan Krios G3i electron microscope operated at an acceleration voltage of 300 kV and equipped with a BioQuantum energy filter and a K3 direct electron detector. Dataset was processed entirely in CryoSPARC[62]. Writing in red highlights failures observed during data processing. **(b)** Schematic data processing workflow for the electron map of Anf(DGK)$_2$ collected with 0.5%

CHAPSO during vitrification. Dataset was collected and processed as described in (a). Writing in red highlights failures observed during data processing. **(c)** Local resolution as calculated by CryoSPARC mapped onto the refined density with different views shown. **(d)** 2.49 Å electron map overlaid with catalytic core of Fe nitrogenase structure (PDB: 8OIE, Anf(DGK)$_2$). Zoomed view highlights lack of- or weak density surrounding AnfG subunits, α-III domains, and FeFecos. Red writing indicates findings during processing/analysis.

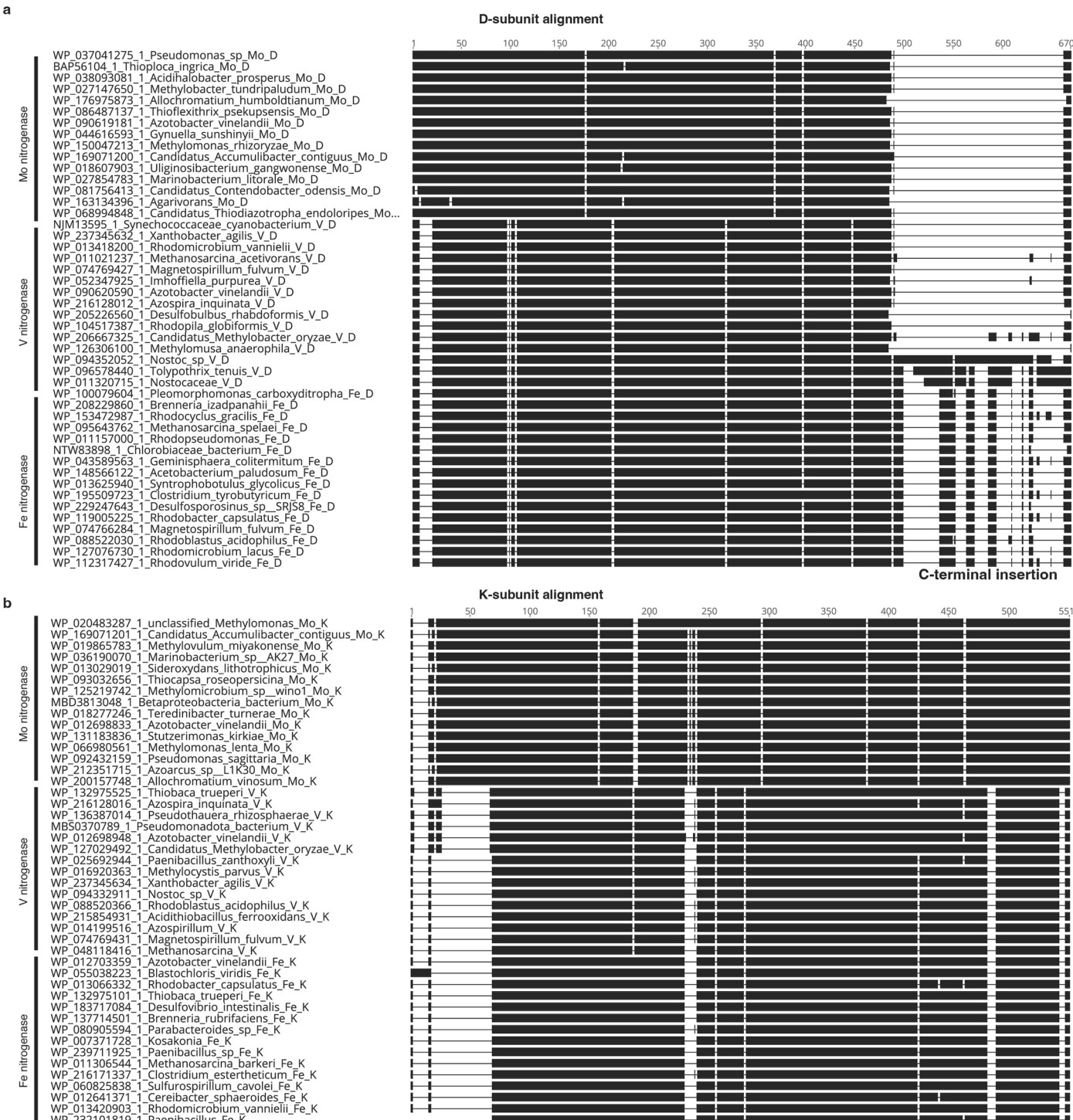

**Extended Data Fig. 6 | Alignments of nitrogenase D and K subunits across species. (a)** Alignment of 46 D-subunits of representative sequence clusters (15 Mo nitrogenases, 15 V nitrogenases, and 16 Fe nitrogenases). Presence/absence of amino acids is shown in black and white, respectively. C-terminal insertion in Fe nitrogenase and select V nitrogenase sequences (cyanobacterial V nitrogenases) is highlighted. C-terminal insertion in cyanobacterial V nitrogenases is divergent of that from Fe nitrogenase in length and sequence. Amino acid sequences were aligned using MUSCLE v5[69]. **(b)** Alignment of 45 K-subunits of representative sequence clusters (15 Mo nitrogenases, 15 V nitrogenases, and 16 Fe nitrogenases). Presence/absence of amino acids is shown in black and white, respectively. N-terminal insertion in Mo nitrogenases is highlighted. Amino acid sequences were aligned using MUSCLE v5.

a

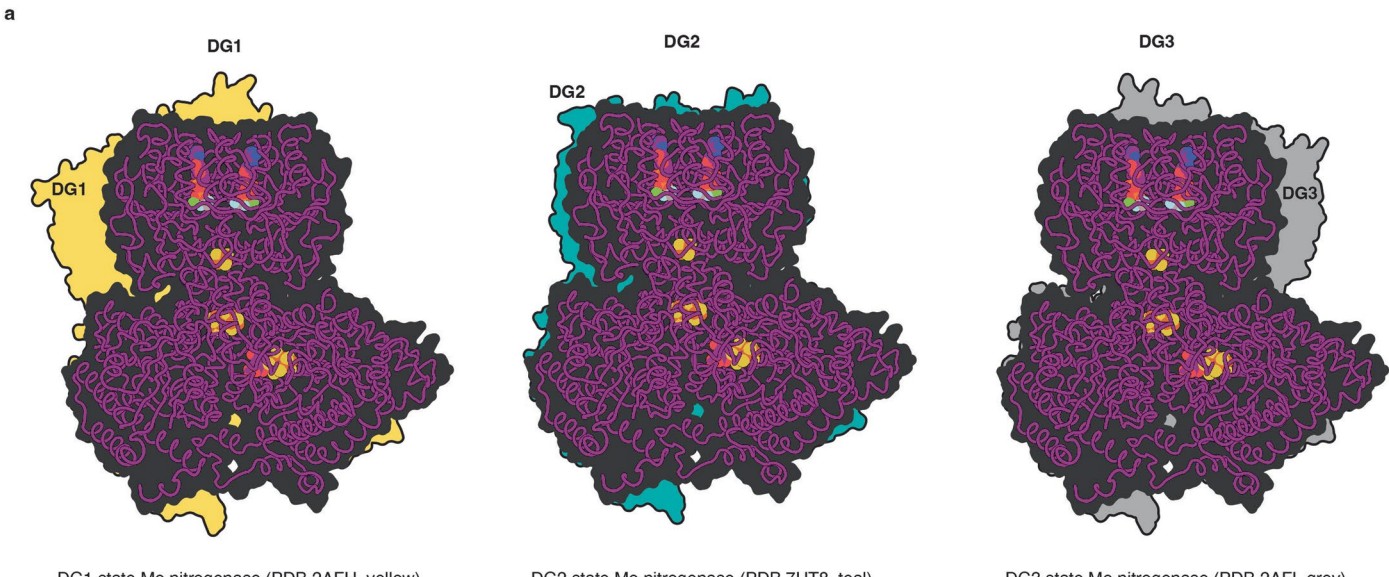

DG1 state Mo nitrogenase (PDB 2AFH, yellow)
aligned to Fe nitrogenase (black/purple)

DG2 state Mo nitrogenase (PDB 7UT8, teal)
aligned to Fe nitrogenase (black/purple)

DG3 state Mo nitrogenase (PDB 2AFI, grey)
aligned to Fe nitrogenase (black/purple)

**Extended Data Fig. 7 | Fe nitrogenase structure docking geometry. (a)** Fe nitrogenase AnfDGKH$_2$ (PDB: 8OIE) overlaid with surface outlines of nitrogenase structures containing varying reductase:catalytic component docking geometries (DG). DG1 in yellow (PDB ID: 2AFH), DG2 in teal (PDB ID: 7UT8), and DG3 in grey (PDB ID: 2AFI). Fe nitrogenase surface outline is shown in black, cartoon is shown in purple, and cofactors are shown as spheres.

**Extended Data Table 1 | Root-mean square deviation between subunits from different nitrogenases**

| D | Mo | V | Fe | K | Mo | V | Fe | G | V | Fe |
|----|------|------|------|----|------|------|------|----|------|------|
| Mo | X | 2.72 | 2.67 | Mo | X | 3.11 | 2.84 | V | X | 1.35 |
| V | 2.72 | X | 1.04 | V | 3.11 | X | 1.23 | Fe | 1.35 | X |
| Fe | 2.67 | 1.04 | X | Fe | 2.84 | 1.23 | X | | | |

*Azotobacter vinelandii* Mo and V nitrogenase (PDB ID 7UTA and 5N6Y, respectively) were aligned to each other and to *Rhodobacter capsulatus* Fe nitrogenase (PDB: 8OIE).

# Reporting Summary

## Statistics

For all statistical analyses, confirm that the following items are present in the figure legend, table legend, main text, or Methods section.

| n/a | Confirmed | |
|---|---|---|
| ☐ | ☒ | The exact sample size (*n*) for each experimental group/condition, given as a discrete number and unit of measurement |
| ☐ | ☒ | A statement on whether measurements were taken from distinct samples or whether the same sample was measured repeatedly |
| ☒ | ☐ | The statistical test(s) used AND whether they are one- or two-sided<br>*Only common tests should be described solely by name; describe more complex techniques in the Methods section.* |
| ☒ | ☐ | A description of all covariates tested |
| ☒ | ☐ | A description of any assumptions or corrections, such as tests of normality and adjustment for multiple comparisons |
| ☐ | ☒ | A full description of the statistical parameters including central tendency (e.g. means) or other basic estimates (e.g. regression coefficient) AND variation (e.g. standard deviation) or associated estimates of uncertainty (e.g. confidence intervals) |
| ☒ | ☐ | For null hypothesis testing, the test statistic (e.g. *F*, *t*, *r*) with confidence intervals, effect sizes, degrees of freedom and *P* value noted<br>*Give P values as exact values whenever suitable.* |
| ☒ | ☐ | For Bayesian analysis, information on the choice of priors and Markov chain Monte Carlo settings |
| ☒ | ☐ | For hierarchical and complex designs, identification of the appropriate level for tests and full reporting of outcomes |
| ☒ | ☐ | Estimates of effect sizes (e.g. Cohen's *d*, Pearson's *r*), indicating how they were calculated |

*Our web collection on statistics for biologists contains articles on many of the points above.*

## Software and code

Policy information about availability of computer code

| Data collection | TotalChrom v. 6.3.4 (PerkinElmer), AcquireMP v. 2.3 (Refeyn Ltd), ICP Expert Software v. 4.1.0 (Agilent), EPU software v. 2.9 - 2.11 (Thermo Scientific) |
|---|---|
| Data analysis | GraphPad Prism 9, DiscoverMP v. 2022 R1 (Refyn Ltd), CryoSPARC 4.1, ChimeraX 1.5, Chimera 1.16, PHENIX 1.21.1, Pymol 2.5, COOT 0.8.9.2, CAVER 3.0.3, AlphaFold 2 (Google DeepMind) |

For manuscripts utilizing custom algorithms or software that are central to the research but not yet described in published literature, software must be made available to editors and reviewers. We strongly encourage code deposition in a community repository (e.g. GitHub). See the Nature Portfolio guidelines for submitting code & software for further information.

## Data

Policy information about availability of data

All manuscripts must include a data availability statement. This statement should provide the following information, where applicable:

- Accession codes, unique identifiers, or web links for publicly available datasets
- A description of any restrictions on data availability
- For clinical datasets or third party data, please ensure that the statement adheres to our policy

All unique materials used in this study are available from the corresponding author upon request. All raw data for mass photometry measurements, kinetic experiments, and protein characterisation are deposited on Edmond, the Open Research Data Repository of the Max Planck Society for public access (DOI: https://

doi.org/10.17617/3.P6SEVC) [67]. The structures reported in this paper are deposited to the Protein Data Bank under the accession codes 8OIE (ADP-AlF3-stabilized Fe nitrogenase complex) and 8PBB (CHAPSO treated partial Fe nitrogenase catalytic component). CryoEM data were deposited to the Electron Microscopy Data Bank under EMD-16890 (ADP-AlF3-stabilized Fe nitrogenase complex) and EMD-17583 (CHAPSO treated partial Fe nitrogenase catalytic component).

The structures used to build our model and compare our structure to are available at the Protein Data Bank under identifiers: 7QQA, 7UTA, 5N6Y. Sequences for the protein alignment were obtained from NCBI non-redundant protein database and are available as table in the source data on EDMOND (DOI: https://doi.org/10.17617/3.P6SEVC).

# Research involving human participants, their data, or biological material

Policy information about studies with human participants or human data. See also policy information about sex, gender (identity/presentation), and sexual orientation and race, ethnicity and racism.

| | |
|---|---|
| Reporting on sex and gender | The research did not involve human participants, their data, or biological material. |
| Reporting on race, ethnicity, or other socially relevant groupings | The research did not involve human participants, their data, or biological material. |
| Population characteristics | The research did not involve human participants, their data, or biological material. |
| Recruitment | The research did not involve human participants, their data, or biological material. |
| Ethics oversight | The research did not involve human participants, their data, or biological material. |

Note that full information on the approval of the study protocol must also be provided in the manuscript.

# Field-specific reporting

Please select the one below that is the best fit for your research. If you are not sure, read the appropriate sections before making your selection.

☒ Life sciences ☐ Behavioural & social sciences ☐ Ecological, evolutionary & environmental sciences

For a reference copy of the document with all sections, see nature.com/documents/nr-reporting-summary-flat.pdf

# Life sciences study design

All studies must disclose on these points even when the disclosure is negative.

| | |
|---|---|
| Sample size | A sample size of n=3 was used for most experiments according to standard scientific practice to validate data quality, repeatability of experiments and reliability of measurement devices. For ICP-OES measurements 2 technical replicates of 2 biological replicates were used. This combination of biological and technical replicates was chosen to ensure robustness and reproducibility of the assay, as well as to validate results on independent days. |
| Data exclusions | No data was excluded. |
| Replication | Most experiments were performed as replicates (n=3) to ensure repeatability and reliability of measurement devices. The number of replications is given in the figure legend for individual experiments and, where applicable, shown as individual data points. Only ICP-OES experiments were performed as a combination of technical and biological replicates to ensure reproducibility on independent days. Single particle cryoEM is based on an average of protein particles within a vitreous layer of ice. Replication is therefore not necessarily required to ensure statistical robustness of structural data. In this work we determined the structure of the iron nitrogenase with and without reductase component with good overlap of core structural features, which can, in essence, be regarded as a biological replicate. |
| Randomization | No randomization was required for the experimental design and workflow of this study. It is of note that stochasticity is introduced during individual data processing steps in CryoSPARC 4.1 (e.g., during 2D classification and during randomization of half map sets during 3D refinement). In the final reconstruction, randomized half sets of particles are used to determine gold-standard Fourier shell correlations at 0.143 level. |
| Blinding | For all studies in this manuscript there was no awareness of group assignment that would have caused biased results. Therefore, blinding was not relevant for data reliability. |

# Reporting for specific materials, systems and methods

We require information from authors about some types of materials, experimental systems and methods used in many studies. Here, indicate whether each material, system or method listed is relevant to your study. If you are not sure if a list item applies to your research, read the appropriate section before selecting a response.

## Materials & experimental systems

| n/a | Involved in the study |
|-----|----------------------|
| ☒ | ☐ Antibodies |
| ☒ | ☐ Eukaryotic cell lines |
| ☒ | ☐ Palaeontology and archaeology |
| ☒ | ☐ Animals and other organisms |
| ☒ | ☐ Clinical data |
| ☒ | ☐ Dual use research of concern |
| ☒ | ☐ Plants |

## Methods

| n/a | Involved in the study |
|-----|----------------------|
| ☒ | ☐ ChIP-seq |
| ☒ | ☐ Flow cytometry |
| ☒ | ☐ MRI-based neuroimaging |

nature portfolio | reporting summary

April 2023

