## [Peer Review File · Nature Structural & Molecular Biology]

Peer Review Information

Manuscript Title: Structural Insights into the Iron Nitrogenase Complex

Corresponding author name(s): Johannes Rebelein

Reviewer Comments & Decisions:

Decision Letter, initial version:
--

Message: 2nd Jun 2023

e:

Dear Dr. Rebelein,

Thank you again for submitting your manuscript "Structural Insights into the Iron Nitrogenase Complex". I apologize for the delay in responding, which resulted from the difficulty in obtaining suitable referee reports. Nevertheless, we now have comments (below) from the 3 reviewers who evaluated your paper. In light of those reports, we remain interested in your study and would like to see your response to the comments of the referees, in the form of a revised manuscript.

You will see that while reviewers appreciate the results, they raise several concerns which will need to be addressed in a revision. Specifically, in line with reviewer's #1 comments, we would encourage you to share the resting state FeFeP structure. Please also revise the methods section with reviewer's #1 guidance.

Please be sure to address/respond to all concerns of the referees in full in a point-by-point response and highlight all changes in the revised manuscript text file. If you have comments that are intended for editors only, please include those in a separate cover letter.

We expect to see your revised manuscript within 6 weeks. If you cannot send it within this time, please contact us to discuss an extension; we would still consider your revision, provided that no similar work has been accepted for publication at NSMB or published

elsewhere.

Reporting Summary:

Please note that all key data shown in the main figures as cropped gels or blots should be presented in uncropped form, with molecular weight markers. These data can be aggregated into a single supplementary figure item. While these data can be displayed in a relatively informal style, they must refer back to the relevant figures. These data should be submitted with the final revision, as source data, prior to acceptance, but you may want to start putting it together at this point.

SOURCE DATA: we request that authors provide, in tabular form, the data underlying the graphical representations used in figures. This is to further increase transparency in data reporting, as detailed in this editorial (<http://www.nature.com/nsmb/journal/v22/n10/full/nsmb.3110.html>). Spreadsheets can be submitted in excel format. Only one (1) file per figure is permitted; thus, for multi-paneled figures, the source data for each panel should be clearly labeled in the Excel file;

alternately the data can be provided as multiple, clearly labeled sheets in an Excel file. When submitting files, the title field should indicate which figure the source data pertains to. We encourage our authors to provide source data at the revision stage, so that they are part of the peer-review process.

Data availability: this journal strongly supports public availability of data. All data used in accepted papers should be available via a public data repository, or alternatively, as Supplementary Information. If data can only be shared on request, please explain why in your Data Availability Statement, and also in the correspondence with your editor. Please note that for some data types, deposition in a public repository is mandatory - more information on our data deposition policies and available repositories can be found below: <https://www.nature.com/nature-research/editorial-policies/reporting-standards#availability-of-data>

Nature Structural & Molecular Biology is committed to improving transparency in authorship. As part of our efforts in this direction, we are now requesting that all authors identified as 'corresponding author' on published papers create and link their Open Researcher and Contributor Identifier (ORCID) with their account on the Manuscript Tracking System (MTS), prior to acceptance. This applies to primary research papers only. ORCID helps the scientific community achieve unambiguous attribution of all scholarly contributions. You can create and link your ORCID from the home page of the MTS by clicking on 'Modify my Springer Nature account'. For more information please visit please visit <http://www.springernature.com/orcid>.

[Redacted]

We look forward to seeing the revised manuscript and thank you for the opportunity to

review your work.

Sincerely,

Katarzyna Ciazynska
(she/her)
Associate Editor
Nature Structural & Molecular Biology
<https://orcid.org/0000-0002-9899-2428>

Reviewers' Comments:

Reviewer #1:

Remarks to the Author:

In this paper, the authors develop a bacterial system for expression and structural analysis of the Fe nitrogenase in *Rhodobacter* species. The work required inserting tags into a previously inaccessible experimental system, and performing anaerobic cryo-EM. They explain that their work contributes a real structure in an area where no real structural work had contributed before. Their paper is written well, and its conclusions will likely be accessible to readers outside of the nitrogenase field. They deserve much credit for creating the needed adjustments to an experimental system, and for describing their results clearly.

Reviewer #2:

Remarks to the Author:

This manuscript by Schmidt et al. describes the first structural characterization of the Fe-only nitrogenase via cryoEM. Impressively, the authors characterized the structure of the entire 10-mer complex inhibited by ADP.AIF to high resolution, in addition to preliminary structural characterization of the catalytic protein on its own. In addition to the structural determination, the authors developed a model system for expressing the Fe-only nitrogenase in the organism *Rhodobacter capsulatus*. The authors provide a thorough analysis of the similarities and differences between the *Rc* Fe-only nitrogenase and *Av* Mo- and V-nitrogenases. I am left wondering why the authors did not characterize the Fe-only nitrogenase from *Av*, since it is perhaps the most popular model organism for studying nitrogenases. However, I don't think that choosing a different organism is detrimental to the manuscript- just a curiosity. The authors provide many reasonable hypotheses as to the function of the AnfG subunit which is not found in the more commonly studied Mo-nitrogenase.

Overall, the manuscript is well written, and the discussion is inciteful. My only major comment regards the data for the structure/data of the resting state FeFeP cryoEM analysis. I also have a handful of minor recommendations, but I do not think that any further experiments need to be carried out. When taken together, the data and discussion of this manuscript

provide a major advance for the nitrogenase community, and I recommend it for publication after minor revisions.

Major comments:

1. If the resting state FeFeP structure was obtained with a high global resolution and is being used in the discussion, the structure should be refined to the extent possible and deposited.

Minor suggestions:

1. Abstract: "Here, we report a 2.35Å structure of the Fe nitrogenase complex from *Rhodobacter capsulatus*..." I think it should be stated in the abstract here that it is the ADP-AIF₃⁻ inhibited structure.
2. Paragraph 2 of the introduction should have references.
3. It is a bit awkward to keep referring to the two protein components simply as "components" rather than proteins.
4. Line 148- You state that there is only ~80% Fe occupancy for the FeFeP based on ICP. This is ambiguous in terms of the number of Fe per FeFeP. Would full occupancy assume that there is no 16th Fe site occupancy? Full occupancy could be 32-34 Fe/protein depending on how you define it.
5. ADP AIF trapped complex formation and characterization:
 - a. Line 153-158: mention that MP was used here. Also, it would be nice to mention the expected masses of the complexes in addition to the measured masses.
 - b. Based on the chromatograms in fig 2c, it looks like there is no 1:1 complex (just 2:1). This is very interesting because it is different from Mo-nitrogenase which forms both the 1:1 and 2:1 upon AIF inhibition. Perhaps including a brief comment on this difference and possible hypothesis as to why they behave differently could be interesting. Perhaps the "distinct structural features" (line 338) may contribute to this.
6. Lines 162-164- I think that this sentence is misleading. It sounds as though the cryoEM structure is of the active complex, and not the inhibited complex.
7. Fig 2a caption: "numbered b, d-e" should say "lettered"
8. Molecular cloning/genetic manipulations methods: The engineered *R.c.* system that you developed for overexpression of Fe-only N₂ase is quite impressive. I think adding a sentence briefly explaining why nitrogenase is not typically recombinantly expressed in *E. coli* would further stress the importance of the expression system that you developed, especially for readers that might not have much of a nitrogenase background. However, I have a few questions that I think can be addressed in the methods section.
 - a. Where was the gentamycin resistance cassette inserted? It says it was used for the interruption of *anfHDGK*, but not specifically which genes are interrupted. I assume that only FeP or FeFeP was interrupted. I would think that both proteins don't need to be interrupted because both proteins are

- being purified with tags, but more information would be relevant. Also, which genes of the Mo-N₂ase were interrupted?
- b. The first paragraph of the genetic manipulation section says that "Subsequently, obtained clones were screened for gentamycin resistance and tetracycline sensitivity..." Is screening for tetracycline sensitivity with *Rc* standard practice? If so, please site, and if not, could you explain why this was done?
 - c. I also have question regarding the tags and affinity purification. Can you please specify which tag was used on which subunit and which terminus and whether a linker was used? Also, with the possibility of lability of Fe in nitrogenase during catalysis, is there concern that the His tag might chelate some Fe from the clusters? Were controls used to compare activity of His-tagged vs not His-tagged protein carried out (or have other publications carried out such controls that you could site)? I know that His tags are sometimes used for purification of other nitrogenases, and I have always thought that using a metal binding tag for nitrogenase is a bit of an odd idea.
9. Growth medium and conditions for protein production methods: Since this organism is not a model organism for studying nitrogenase, could more information be included? What size growths were carried out and approximately how much protein was obtained per L of growth? Genomic expression in *Av* is typically carried out at a large scale since it is not recombinant, so I am curious how that compares to your system.
10. Protein purification/SDS-PAGE methods: I have a few questions/comments on this section:
- a. If I understand correctly, all genes for recombinant expression of Fe-only nitrogenase were expressed together. In the protein purification methods, it states, "The cleared cell extract was then applied to high salt buffer equilibrated HisTrap HP and StrepTactinXT ...". Was the lysate split and then applied, or was it the lysate not split and then applied to the columns sequentially such that both component proteins could be purified from the same lysate?
 - b. Please give the composition of the binding buffer used.
 - c. "AnfH2 eluted in a clear peak..." Just to clarify, this means a single peak, right? Not colorless fractions?
11. Mass photometry methods: Please specify what proteins/molecular masses were included in the "in-house purified protein mixture" used for calibration.
12. Extended data Fig 4: The overlays are hard to understand. Could you find a way to make them more clear? Maybe making them slightly transparent (all panels) and using a cartoon representation that is less thick (panels b and d) would help. Which cryo structure was used for panel e, one of the complexes or resting-state MoFeP? Main text states PDB ID7UTA. Include PDB IDs in the figure caption, along with which structure they represent (same comment regarding main text fig 4).

Reviewer #3:
Remarks to the Author:

Rebelein and coworkers are the first to structurally characterize the Fe nitrogenase. Using cryo-EM, they solved the Fe nitrogenase structure revealing the architecture and location of an additional subunit (the G-subunit) that is only found in the alternative forms of nitrogenase. The structure is used to hypothesize roles for this subunit. Their structure further showed that the Fe cluster strongly resembles FeMoCo from the Mo nitrogenase, which is a very important finding. The methodology appears sound. I appreciated that authors showed us a number of views of the maps. Authors do a nice job of comparing and contrasting nitrogenase structures. I would strongly recommend publication as this is an important structure and a well written paper.

minor

1. Typos or missing words that should be included (highlighted in blue parts are the changes)
 - a. Line 24: 2.35-Å **resolution** structure
Also, this is the only time the authors put a "-" between the number and the unit when referring to resolution. The rest of the document does not have the "-"
 - b. Line 49: The homodimeric reductase component contains **an** [Fe₄S₄]-cluster
 - c. Line 161: 2.35 Å **resolution** map
 - d. Line 193: **an** [Fe₈S₇]-cluster
 - e. Line 240: hydrogen bonding (**H-bonding**) between
 - f. Line 267: α -helical subunit: VnfG and AnfG (**the G-subunit**)
 - g. Line 403: halves **to** "communicate"
2. Referencing appears appropriate. I only noticed one to add: Krahn, E., Weiss, B., Kröcher, S. et al. The Fe-only nitrogenase from *Rhodobacter capsulatus*: identification of the cofactor, an unusual, high-nuclearity iron-sulfur cluster, by Fe K-edge EXAFS and ⁵⁷Fe Mössbauer spectroscopy. *J Biol Inorg Chem* **7**, 37–45 (2002).

Author Rebuttal to Initial comments

Point-by-Point Reply

General changes:

Due to the publication of the X-ray structure of the FeFe protein (the Anf(DGK)₂) published during the review process by Einsle and coworkers in *Nature Catalysis* (<https://doi.org/10.1038/s41929-023-00952-1>), we have introduced the following changes:

The structures of the FeMoco, the FeVco and just recently the FeFeco have been structurally confirmed by X-Ray crystallography [Lit.].

~~The structure of the FeFeco has not been experimentally verified so far, leaving room for debate whether it fits the general scheme described above.~~

To gain molecular insights into the differences among the three nitrogenase isoenzymes, we set out to solve the first structure of the entire Fe nitrogenase complex, the first complex of any alternative, V or Fe nitrogenase. ~~the nitrogenase isozyme for which no structure has been published, yet.~~

Referee 1:

We thank the reviewer for the positive feedback, specifically for highlighting the accessibility of our paper to a broader audience and for credit regarding the findings and their presentation.

Referee 2:

We thank Referee 2 for the detailed review and positive assessment of our work.

Major comments:

- 1) If the resting state FeFeP structure was obtained with a high global resolution and is being used in the discussion, the structure should be refined to the extent possible and deposited.

Reply: We refined the FeFe protein structure to the extent possible and deposited the resulting structure to the PDB: 8PBB and the electron density map at the Electron Microscopy Data Bank: EMD-17583. The structure, the electron density map and the validation report are available for review here: <https://owncloud.gwdg.de/index.php/s/XeJSi6jLF4zsYuu> password:

NitrOgenaseFOrReview

The materials and methods, as well as supplementary tables (i.e. cryoEM refinement data tables) have been updated accordingly. The following sentence is the text was modified:

“Intriguingly, we were not able to resolve electron densities for parts of AnfD (25% of the residues) and AnfK (4% of the residues) in the AnfG-free complex, including the αIII domain and the FeFeco and parts of the P-cluster and its binding site (PDB: 8PBB).”

Minor suggestions:

- 1) Abstract: “Here, we report a 2.35Å structure of the Fe nitrogenase complex from *Rhodobacter capsulatus*...” I think it should be stated in the abstract here that it is the ADP-AIF₃-inhibited structure.

Reply: Thank you for the suggestion, the abstract now states:

“Here, we report a 2.35 Å structure of the ADP-AIF₃-stabilized Fe nitrogenase complex.”.

2) Paragraph 2 of the introduction should have references.

Reply: Thank you for this comment we have added the following citations to the paragraph:

9. Eady, R.R., *Structure–Function Relationships of Alternative Nitrogenases*. Chemical Reviews, 1996. **96**(7): p. 3013-3030.
10. Jasniewski, A.J., et al., *Reactivity, Mechanism, and Assembly of the Alternative Nitrogenases*. Chemical Reviews, 2020. **120**(12): p. 5107-5157.
11. Ribbe, M.W., et al., *Biosynthesis of Nitrogenase Metalloclusters*. Chemical Reviews, 2014. **114**(8): p. 4063-4080.

3) It is a bit awkward to keep referring to the two protein components simply as “components” rather than proteins.

Reply: We adapted our wording slightly. We now initially mention the individual proteins as the “...the reductase **component protein** (NifH₂, VnfH₂, AnfH₂) and the catalytic **component protein** (Nif(DK)₂, Vnf(DGK)₂, Anf(DGK)₂).”

Thereafter we would prefer to keep using “components”. This is in line with previous literature, which refers to the nitrogenase proteins as components (often after initially referring to them as component proteins) e.g.:

<https://pubs.acs.org/doi/full/10.1021/acs.chemrev.0c00067>

<https://link.springer.com/article/10.1007/s00775-014-1225-3>

<https://www.sciencedirect.com/science/article/pii/S0162013421003378>

<https://www.nature.com/articles/249805a0>

<https://www.nature.com/articles/ncomms11426>

[10.1021/acs.chemrev.9b00663](https://pubs.acs.org/doi/full/10.1021/acs.chemrev.9b00663)

We think this nomenclature reduces redundancy and thus makes the article more readable.

However, we don't feel strongly about this and are happy to adapt the nomenclature further, if wished for by the referee/editor.

4) Line 148- You state that there is only ~80% Fe occupancy for the FeFeP based on ICP. This is ambiguous in terms of the number of Fe per FeFeP. Would full occupancy assume that there is no 16th Fe site occupancy? Full occupancy could be 32-34 Fe/protein depending on how you define it.

Reply: We added a statement indicating that Fe occupancy was calculated with 32 Fe atoms per catalytic component. We assumed full occupancy to include the 16th Fe site:

“Metal quantification via inductively coupled plasma optical emission spectroscopy (ICP-OES) suggested full Fe occupancy for the reductase component and ~80% occupancy for the catalytic component (**at 32 expected Fe atoms per catalytic component**, Extended Data Fig. 2).”

5) ADP AIF trapped complex formation and characterization:

- a. Line 153-158: mention that MP was used here. Also, it would be nice to mention the expected masses of the complexes in addition to the measured masses.

Reply: We added the following statement:

“Following size exclusion chromatography (SEC), we detected a protein complex of ~360 kDa in size **via mass photometry** (Fig. 1c, bottom). The measured masses of the individual nitrogenase components were 236 kDa for Anf(DGK)₂ and 69 kDa for AnfH₂ (Fig. 1c, top and middle), thus indicating an Anf(DGK)₂(H₂)₂ stoichiometry of the complex (**expected mass 374 kDa**).”

- b. Based on the chromatograms in fig 2c, it looks like there is no 1:1 complex (just 2:1). This is very interesting because it is different from Mo-nitrogenase which forms both the 1:1 and

2:1 upon AIF inhibition. Perhaps including a brief comment on this difference and possible hypothesis as to why they behave differently could be interesting. Perhaps the “distinct structural features” (line 338) may contribute to this.

Reply: Thank you for this important suggestion. We have added the following statement:

Furthermore, the G-subunit might be involved in stabilizing the interaction between reductase and catalytic component. For the Fe nitrogenase we only observed ADP-AIF₃ trapped complexes consisting of two reductase and one catalytic component during SEC analysis (see Fig 2c). This contrasts with the Mo nitrogenase, which forms in the presence of beryllium fluoride mainly complexes consisting of one reductase and one catalytic component [30].

6) Lines 162-164- I think that this sentence is misleading. It sounds as though the cryoEM structure is of the active complex, and not the inhibited complex.

Reply: We agree that this could be misinterpreted. To circumvent this, we specify that we solved the structure of the “...ADP-AIF₃-stabilized, AnFH₂-bound complex.”

7) Fig 2a caption: “numbered b, d-e” should say “lettered”

Reply: Changed accordingly.

8) Molecular cloning/genetic manipulations methods: The engineered R.c. system that you developed for overexpression of Fe-only N₂ase is quite impressive. I think adding a sentence briefly explaining why nitrogenase is not typically recombinantly expressed in E coli would further stress the importance of the expression system that you developed, especially for readers that might not have much of a nitrogenase background.

Reply: We thank the reviewer for this suggestion and have added the following statement:

Despite recent advances toward expressing complex heterometallic proteins in *Escherichia coli*, it remains exceedingly challenging to heterologously express nitrogenases with completely assembled metalloclusters in *E. coli*. Therefore, we used here the natural diazotroph *R. capsulatus*, which is genetically more accessible than *A. vinelandii*.

However, I have a few questions that I think can be addressed in the methods section.

a. Where was the gentamycin resistance cassette inserted? It says it was used for the interruption of *anfHDGK*, but not specifically which genes are interrupted. I assume that only FeP or FeFeP was interrupted. I would think that both proteins don't need to be interrupted because both proteins are being purified with tags, but more information would be relevant. Also, which genes of the Mo-N₂ase were interrupted?

Reply: The gentamycin resistance cassette replaced parts of *anfH*, the entire *anfD*, *anfG* and parts of *anfK* from the *R. capsulatus* genome. Extended Data Fig. 1 shows the approximate position of this insertion. The same figure also visualises the position of the 3.9 kb deletion inside the *nif* operon, including *nifH*, *nifD* and *nifK*. We agree that the deletion of the *anf* operon would not have been necessary for purification purposes but it aided the development of the purification strain. For example, the deletion allowed us to test *in vivo* the activity of the the affinity tagged Fe nitrogenase. References to the Extended Figure 1 were included in the Results section: Engineering *Rhodobacter capsulatus* for nitrogenase expression:

“In summary, we introduced five genetic modifications into *R. capsulatus* (depicted in Extended Data Fig. 1) that render the purple non-sulphur bacterium an ideal platform for the plasmid-based production and characterisation of the Fe nitrogenase.”

And the Methods section:

“For the deletion of *anfHDGK*, a gentamycin resistance cassette was inserted into the *anfHDGK* locus, thereby interrupting the operon (Extended Data Fig. 1).”

- b. The first paragraph of the genetic manipulation section says that “Subsequently, obtained clones were screened for gentamycin resistance and tetracycline sensitivity...” Is screening for tetracycline sensitivity with Rc standard practice? If so, please site, and if not, could you explain why this was done?

Reply: As described in table S2, plasmid pBS85-Bsal-genR is a suicide vector that contains a gentamycin resistance gene flanked by the homologous recombination sites for genomic integration of the plasmid and a tetracycline resistance gene in the backbone. Following the first recombination event, clones that integrated pBS85-Bsal-genR into their genome were expected to be both gentamycin and tetracycline resistant. However, we decided to only select for gentamycin resistance in the first step. Following the second recombination event (removal of the pBS85-Bsal-genR backbone), the respective clones were expected to lose their tetracycline resistance but still be gentamycin resistant. Thus, in the second step clones were screened for gentamycin resistance and tetracycline sensitivity. The applied dosage of tetracycline was chosen according to <https://doi.org/10.1073/pnas.97.2.859>. Trying to clarify this, we have modified the respective statement to:

“Subsequently, double recombinant clones were identified through screening for gentamycin resistance and tetracycline sensitivity (resulting from the loss of the suicide vector backbone through double recombination) on peptone yeast (PY) agar plates...”

- c. I also have question regarding the tags and affinity purification. Can you please specify which tag was used on which subunit and which terminus and whether a linker was used? Also, with the possibility of lability of Fe in nitrogenase during catalysis, is there concern that the His tag might chelate some Fe from the clusters? Were controls used to compare activity of His-His-tagged vs not His-tagged protein carried out (or have other publications carried out such controls that you could site)? I know that His tags are sometimes used for purification of other nitrogenases, and I have always thought that using a metal binding tag for nitrogenase is a bit of an odd idea.

Reply: Thank you for the attentive comment. The position and nature of the tags is described in line 124 - 125: “...and fused a His₆-tag to the AnfH N-terminus and a Strep-tag II to the C-terminus of AnfD..” To clarify this, we have also updated the description of how the expression plasmid was constructed at the end of the “Molecular cloning” section:

“...using primers oMM0223 plus oMM0224 for the insertion of the Strep-tag II at the AnfD C-terminus and oMM0510 plus oMM0511 for the insertion of the (His)₆-tag at the AnfH N-terminus. Both tags were inserted without any linker sequence.”

We agree that using a metal binding tag might be concerning when working with a metalloenzyme. However, His-tags have been successfully used for the purification of various nitrogenase proteins including, but not limited NifH, NifDK, VnfDGK, AnfDGK and NifEN. For more details, please see the publications below:

<https://doi.org/10.1126/science.aaz6748>

<https://doi.org/10.1073/pnas.0904408106>

<https://doi.org/10.1073/pnas.1102773108>

<https://doi.org/10.1021/bi981165b>

<https://doi.org/10.1038/s41564-017-0091-5>

<https://doi.org/10.1038/ncomms11426>

Furthermore, our tagged Fe nitrogenase matches the *in vitro* nitrogen fixing activity that is observed with a non-tagged Fe nitrogenase from the same organism (Schneider, K., et al., 1997). We have updated the manuscript including this information:

“*In vitro*, the Fe nitrogenase converted dinitrogen (N₂) to ammonia (NH₃) at a maximal rate of 0.69 nmol × nmol (Anf(DGK)₂)⁻¹ × s⁻¹, closely matching the previously published value of 0.72 nmol × nmol (Anf(DGK)₂)⁻¹ × s⁻¹ that was observed with an untagged Fe nitrogenase from *R. capsulatus* [37].”

- 9) Growth medium and conditions for protein production methods: Since this organism is not a model organism for studying nitrogenase, could more information be included? What size growths were carried out and approximately how much protein was obtained per L of growth? Genomic expression in Av is typically carried out at a large scale since it is not recombinant, so I am curious how that compares to your system.

Reply: Thank you for the comment. *R. capsulatus* was cultivated in 800 mL of RCV medium for protein purification. We have updated a statement in the section “Growth medium and conditions for protein production”

“Subsequently, 800 mL RCV medium was inoculated with an optical density of 0.1 at 660 nm (OD₆₆₀) for protein production.”

To clarify the yields, we added the following statement at the end of the “Protein Purification” section:

“Following the described protein purification procedure, approximately 3.5 mg of catalytic and 12 mg of reductase component were obtained per litre of *R. capsulatus* growth culture”

- 10) Protein purification/SDS-PAGE methods: I have a few questions/comments on this section:
- a. If I understand correctly, all genes for recombinant expression of Fe-only nitrogenase were expressed together. In the protein purification methods, it states, “The cleared cell extract was then applied to high salt buffer equilibrated HisTrap HP and Strep-TactinXT ...”. Was the lysate split and then applied, or was it the lysate not split and then applied to the columns sequentially such that both component proteins could be purified from the same lysate?

Reply: The entire lysate was applied sequentially to the two columns that were stacked on top of each other. This way, the lysate would first get in contact with the HisTrap™ HP column (where the His-tagged reductase component would bind) and then with the Strep-Tactin®XT column (where the catalytic component would bind). Thus, both components could be purified from the same lysate. To clarify this the manuscript was adapted as follows:

“The entire cleared cell extract was then applied sequentially to high salt buffer equilibrated HisTrap™ HP (Cytiva Europe GmbH, Freiburg, Germany) and Strep-Tactin®XT 4Flow® high capacity (IBA Lifesciences, Göttingen, Germany) columns via a ÄKTA pure™ chromatography system (Cytiva Europe GmbH, Freiburg, Germany).”

- b. Please give the composition of the binding buffer used.

Reply: We thank the reviewer for catching this wording mistake. We meant to refer to the buffer used as “high salt buffer” instead of “binding buffer”. As described in the text “high salt buffer” was composed of 50 mM TRIS (pH = 7.8), 500 mM NaCl, 10% glycerol and 4 mM sodium dithionite. We updated the respective sentence to:

“After extensive washing with high salt buffer, the catalytic component was eluted from the Strep-Tactin®XT column with high salt buffer supplemented with 50 mM biotin.”

- c. “AnfH2 eluted in a clear peak...” Just to clarify, this means a single peak, right? Not colorless fractions?

Reply: This indeed means as a single peak. To avoid confusion, we now state:

“AnfH₂ eluted in a single peak at around 205 mL and was subsequently concentrated”

- 11) Mass photometry methods: Please specify what proteins/molecular masses were included in the “in-house purified protein mixture” used for calibration.

Reply: We adapted the statement to:

“MP contrast was calibrated to molecular masses using 50 nM of a mixture of citrate synthase complexes with varying complex stoichiometries of masses ranging from 86 kDa to 430 kDa.”

Any protein with a known stable complex mass can be used to calibrate mass photometry contrasts (we previously used commercially available SEC standards). The current mixture of citrate synthase complexes is pending for patent approval.

- 12) Extended data Fig 4: The overlays are hard to understand. Could you find a way to make them more clear? Maybe making them slightly transparent (all panels) and using a cartoon representation that is less thick (panels b and d) would help. Which cryo structure was used for panel e, one of the complexes or resting-state MoFeP? Main text states PDB ID7UTA. Include PDB IDs in the figure caption, along with which structure they represent (same comment regarding main text fig 4).

Reply: We updated Extended Data Fig. 4 to make the overlays easier to understand and make the data clearer.

The lower part of panel **b** was updated to show the flexible loop over the alpha III domain more clearly. The unresolved region in the Mo nitrogenase structure 7UTA is now outlined in black. We worked with transparency of the remaining structure to reduce complexity. The upper part of panel **b** was left unchanged as it serves as an overview to identify the region of interest for the lower part of panel **b**.

Panel **d** was updated in a similar fashion to **b**. We now highlight the structural features of interest by outlining them and use transparency in the remaining structure. We furthermore added a box to highlight what region of nitrogenase is depicted.

Figure **e** was split to show the cartoon putty representations to avoid the overlay. Overlaying the three individual structures – even when using transparency – always obscured/hid information. We hope that the extended data figure 4 is now easier to read/interpret.

We updated the Figure legends to include information of PDB ID and which structures they represent. We now additionally point out the PDB IDs/structure states in Extended Data Figure 4 itself.

Referee 3:

We thank the reviewer for the positive assessment of our work and the strong recommendation for publication. We are equally grateful for the thorough read-through and suggested adjustments.

1. Typos or missing words that should be included

a. Line 24: 2.35-Å resolution structure

Also, this is the only time the authors put a “-” between the number and the symbol when referring to resolution. The rest of the document does not have the “_”

Reply: We removed the hyphen in the abstract.

b. Line 49: The homodimeric reductase component contains an [Fe4S4]-cluster

Reply: Adapted.

c. Line 161: 2.35 Å resolution map

Reply: Adapted.

d. Line 193: an [Fe8S7]-cluster

Reply: Adapted.

e. Line 240: hydrogen bonding (H-bonding) between

Reply: Adapted.

f. Line 267: α -helical subunit: VnfG and AnfG (the G-subunit)

Reply: We agree that this helps understanding and added the G-subunit parentheses.

g. Line 403: halves to “communicate”

Reply: Adapted.

2. Referencing appears appropriate. I only noticed one to add: Krahn, E., Weiss, B., Kröckel, M. et al. The Fe-only nitrogenase from *Rhodobacter capsulatus*: identification of the cofactor, an unusual, high-nuclearity iron-sulfur cluster, by Fe K-edge EXAFS and ⁵⁷Fe Mössbauer spectroscopy. *J Biol Inorg Chem* 7, 37–45 (2002).

Reply:

Indeed, this reference was overlooked, we included it in the following statement:

“As previously proposed [16, 45], the FeFeco is a [Fe₈S₉C-(*R*)-homocitrate]-cluster (Fig. 2a and f) that in contrast to the Mo and V nitrogenases contains no transition metal other than iron (based on ICP-OES).”

Decision Letter, first revision:

Message: Our ref: NSMB-A47533A

13th Jul 2023

Dear Dr. Rebelein,

Thank you for submitting your revised manuscript "Structural Insights into the Iron Nitrogenase Complex" (NSMB-A47533A). It has now been seen by the original referees and their comments are below. The reviewers find that the paper has improved in revision, and therefore we'll be happy in principle to publish it in Nature Structural & Molecular Biology, pending minor revisions to satisfy the referees' final requests and to comply with our editorial and formatting guidelines.

Sincerely,

Katarzyna Ciazynska
(she/her)
Associate Editor
Nature Structural & Molecular Biology
<https://orcid.org/0000-0002-9899-2428>

Reviewer #2 (Remarks to the Author):

The authors addressed all of my concerns, and I recommend that the article is published.

Reviewer #3 (Remarks to the Author):

Authors have addressed all of my concerns. The paper is ready for publication.

Final Decision Letter:

Message 12th Sep 2023

:

Dear Dr. Rebelein,

We are now happy to accept your revised paper "Structural Insights into the Iron Nitrogenase Complex" for publication as an Article in Nature Structural & Molecular Biology.

As soon as your article is published, you can generate your shareable link by entering the DOI of your article here: http://authors.springernature.com/share. Corresponding authors will also receive an automated email with the shareable link

Your paper will be published online soon after we receive proof corrections and will appear in print in the next available issue. You can find out your date of online publication by contacting the production team shortly after sending your proof corrections. Content is published online weekly on Mondays and Thursdays, and the embargo is set at 16:00 London time (GMT)/11:00 am US Eastern time (EST) on the day of publication. Now is the time to inform your Public Relations or Press Office about your paper, as they might be

interested in promoting its publication. This will allow them time to prepare an accurate and satisfactory press release. Include your manuscript tracking number (NSMB-A47533B) and our journal name, which they will need when they contact our press office.

About one week before your paper is published online, we shall be distributing a press release to news organizations worldwide, which may very well include details of your work. We are happy for your institution or funding agency to prepare its own press release, but it must mention the embargo date and Nature Structural & Molecular Biology. If you or your Press Office have any enquiries in the meantime, please contact press@nature.com.

Please note that *Nature Structural & Molecular Biology* is a Transformative Journal (TJ). Authors may publish their research with us through the traditional subscription access route or make their paper immediately open access through payment of an article-processing charge (APC). Authors will not be required to make a final decision about access to their article until it has been accepted. <https://www.springernature.com/gp/open-research/transformative-journals> Find out more about Transformative Journals

Sincerely,

Katarzyna Ciazynska
(she/her)
Associate Editor
Nature Structural & Molecular Biology
<https://orcid.org/0000-0002-9899-2428>
